# How to Move Your Dragon:
# Text-to-Motion Synthesis for Large-Vocabulary Objects

Wonkwang Lee [1 2]   Jongwon Jeong [* 2]   Taehong Moon [* 2]
Hyeon-Jong Kim [2]   Jaehyeon Kim [† 3]   Gunhee Kim [1]   Byeong-Uk Lee [2]

## Abstract

Motion synthesis for diverse object categories holds great potential for 3D content creation but remains underexplored due to two key challenges: (1) the lack of comprehensive motion datasets that include a wide range of high-quality motions and annotations, and (2) the absence of methods capable of handling heterogeneous skeletal templates from diverse objects. To address these challenges, we contribute the following: First, we augment the Truebones Zoo dataset—a high-quality animal motion dataset covering over 70 species—by annotating it with detailed text descriptions, making it suitable for text-based motion synthesis. Second, we introduce rig augmentation techniques that generate diverse motion data while preserving consistent dynamics, enabling models to adapt to various skeletal configurations. Finally, we redesign existing motion diffusion models to dynamically adapt to arbitrary skeletal templates, enabling motion synthesis for a diverse range of objects with varying structures. Experiments show that our method learns to generate high-fidelity motions from textual descriptions for diverse and even unseen objects, setting a strong foundation for motion synthesis across diverse object categories and skeletal templates. Qualitative results are available on this link.

## 1. Introduction

Imagine a world where creating lifelike motions for any creature or object is as effortless as describing it in words—a world where the majestic flight of a dragon or the intricate crawl of a centipede can be seamlessly brought to life with the power of intuitive text-based synthesis. The ability to create realistic 3D motions for such a diverse array of entities has long been a shared aspiration across creative communities and industries, unlocking profound potential for animation, gaming, virtual reality, and beyond. However, achieving such realism has traditionally required significant manual effort and expertise, making the process both labor-intensive and time-consuming (Baran & Popović, 2007).

Despite advancements in data-driven motion synthesis, two key challenges hinder progress in generating motions for diverse objects. First, the lack of high-quality motion datasets (Kapon et al., 2024) spanning a broad spectrum of objects with diverse skeletal structures and rich annotations limits the opportunity to explore text-driven motion synthesis for large-vocabulary objects. Second, existing methods rely heavily on fixed skeletal rig templates, such as those designed for humans (Tevet et al., 2023; Guo et al., 2024), making them ill-suited for handling the heterogeneous skeletal configurations found in real-world 3D motion data in a unified manner. From the quadrupedal stance of a horse to the winged anatomy of a bird or the fantastical form of a dragon, synthesizing coherent and realistic motions across such diverse skeletal structures remains an open challenge.

To address these challenges, we propose a novel framework for text-to-motion synthesis that enables motion generation across arbitrary skeletal structures, accommodating the diversity of real-world object motion data.

Specifically, we make the following contributions:

- **Novel Problem Setup**: To the best of our knowledge, this is the first work to tackle text-driven motion synthesis for a broad range of objects with significantly different skeletal structures within a unified framework.

- **High-Quality Text Descriptions**: We curate a comprehensive set of human-labeled descriptions for Truebones Zoo dataset (Truebones, 2022), covering motions over 70 species with unique skeletal templates.

- **Rig Augmentation**: We introduce novel rig augmentation methods—adjusting bone lengths/numbers and rest poses—to enhance the model's adaptability and generalization to diverse skeletal templates.

---

[*]Equal contribution. [†]Work done at KRAFTON. [1]Seoul National University [2]KRAFTON [3]NVIDIA. Correspondence to: Byeong-Uk Lee <lview94@gmail.com>.

*Proceedings of the 42nd International Conference on Machine Learning*, Vancouver, Canada. PMLR 267, 2025. Copyright 2025 by the author(s).

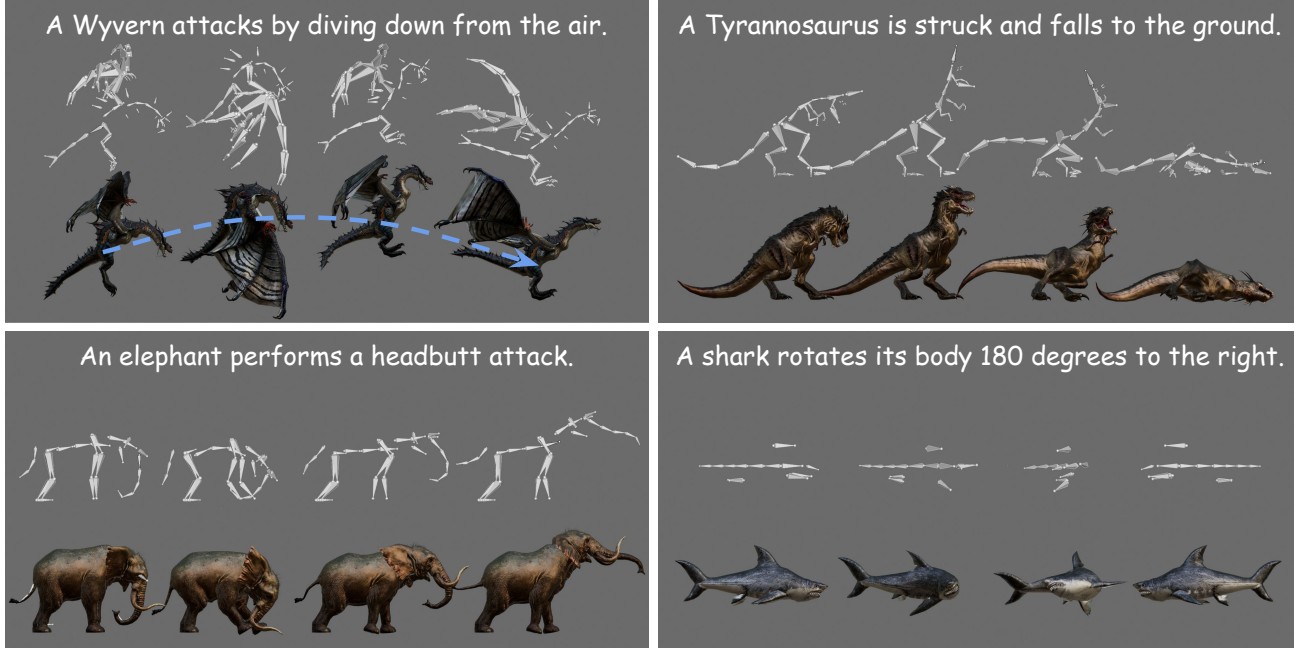

*Figure 1.* In this work, we address the task of synthesizing high-fidelity 3D motions for a broad spectrum of objects with diverse skeletal structures, characterized by varying joint counts and differing dependencies, including animals, dinosaurs, and imaginary creatures, based on textual descriptions. The synthesized motions can be applied to rigged meshes to generate visually appealing 3D animations.

- **Generalized Motion Diffusion Models**: We extend motion diffusion models (Tevet et al., 2023), originally constrained to a single fixed skeletal template, by incorporating tree positional encoding (TreePE) (Shiv & Quirk, 2019) and rest pose encoding (RestPE) , allowing dynamic adaptation to diverse joint hierarchies and skeletal structures.

Extensive experiments on Truebones Zoo dataset demonstrate our framework's ability to generate high-fidelity motions conditioned on textual descriptions, or even synthesize motions for novel objects downloaded from the web.

To inspire future work and further advancements, we will release the code for our data and model pipelines, along with the annotated captions, establishing a comprehensive benchmark for motion synthesis across diverse objects with heterogeneous skeletal structures.

## 2. Related Work

### 2.1. Human Motion Synthesis

Most research in motion synthesis has focused on human motion with a fixed skeletal template, forming the foundation of progress in this field. Early work emphasized text-driven motion synthesis (Plappert et al., 2016; Guo et al., 2022), generating human motion from textual descriptions. Recent work introduced fine-grained control (Zhang et al., 2023) for specific body parts, open-vocabulary motion generation (Liang et al., 2024) for arbitrary text inputs, and

skeleton-agnostic designs for humans and anthropomorphic biped characters (Zhang et al., 2024). Despite successes, most approaches either rely on fixed skeletal templates or exclusively focus on human or anthropomorphic motion, limiting their applicability to more diverse objects with heterogeneous anatomical structures and motion patterns.

### 2.2. Non-Human Motion Synthesis

Non-human motion synthesis has received comparatively less attention, though recent efforts have started addressing this gap. SinMDM (Raab et al., 2024) identified internal motion motifs within an arbitrary object motion, enabling the generation of diverse, flexible-length motions that preserve structural consistency. OmniMotion-GPT (Yang et al., 2024) leveraged human motion data to synthesize quadruped animal motions, demonstrating the transferability of human motion to non-human domains. Similarly, Zhao et al. (2024) introduced a learning-based retargeting method capable of adapting dog motions to a T-rex, horse, and hamster, though it requires training separate CycleGANs (Zhu et al., 2017) for each source-target pair. MAS (Kapon et al., 2024) extended a 2D motion diffusion model with a 2D-to-3D lifting mechanism to synthesize 3D equine motions. However, all these approaches rely on fixed skeletal templates or object-specific components and designs, limiting their adaptability to arbitrary structures across a large vocabulary of objects.

In contrast, CharacterMixer (Zhan et al., 2024) and Skin-Mixer (Nuvoli et al., 2022) propose rig-blending techniques

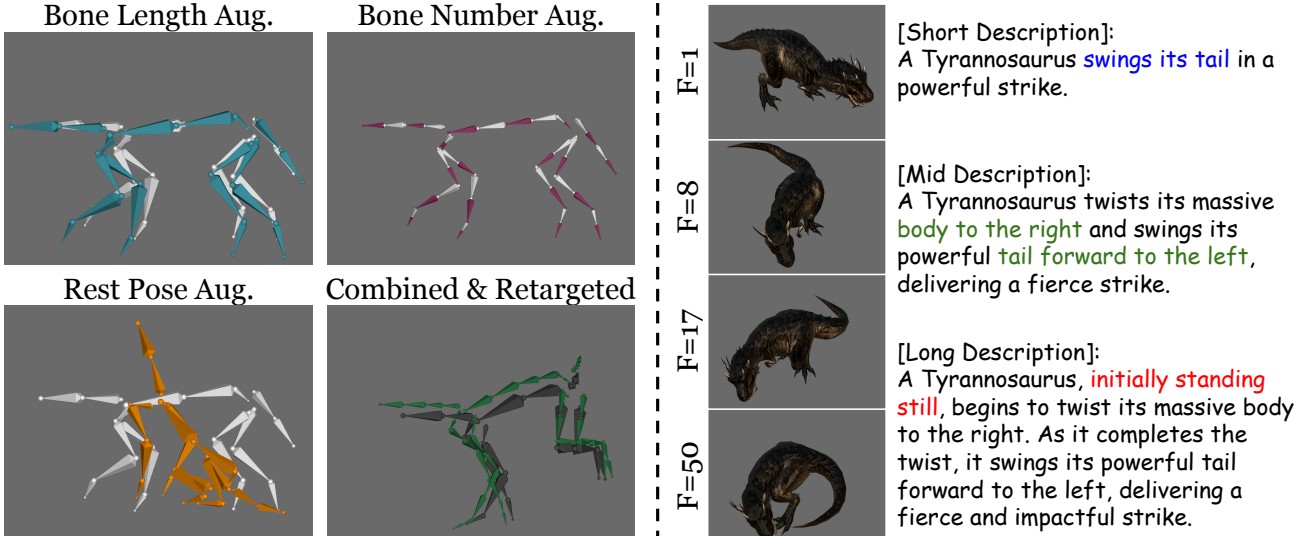

*Figure 2.* **Left**: Given the original skeleton template (white), the rig augmentation technique adjusts bone length (cyan), quantity (pink), rest poses (orange), or their combinations. The resulting skeleton template is then retargeted (green) to the original poses (dark) at each frame to generate data that maintains the original motion dynamics while incorporating diverse skeletal templates. **Right**: Our annotated captions accurately capture motion, delivering different levels of detail through short, mid, and long descriptions. Details are added gradually, ranging from high-level actions (blue) to part-level dynamics (green) and initial postures (red).

that create hybrid characters and their motions constructed from multiple heterogeneous skeletons. While these methods demonstrate the potential to generalize motion across structurally diverse rigs, their reliance on motion retargeting limits flexibility in generating novel or diverse motions.

## 3. Background: Hierarchical Skeletal Rig

A hierarchical skeletal rig forms the foundation for representing 3D motion. It encodes an object's motion as sequence of 3D poses $\mathbf{P}_{\text{global}} \in \mathbb{R}^{F \times J \times 3}$, where $F$ is the number of frames and $J$ represents the number of joints. These 3D poses are reconstructed by integrating static features, which define the skeleton's topology and configuration, with dynamic features, which capture the temporal aspects of motion across frames.

**Static Features** include a tree hierarchy of $J$ joints ($\mathcal{S}$) that represents a skeletal topology, and the rest pose ($\mathbf{P}_{\text{rest}} \in \mathbb{R}^{J \times 3}$) that encodes joint offsets relative to their parents. Each joint $j$ is associated with a parent joint $\mathcal{P}(j)$.

**Dynamic Features** describe motion via a sequence of joint translations ($\mathbf{p} \in \mathbb{R}^{F \times J \times 3}$) and rotations ($\mathbf{R} \in \mathbb{R}^{F \times J \times D}$), where $F$ is the number of frames and $D$ the dimension of the rotation. $\mathbf{p}[f, j]$ and $\mathbf{R}[f, j]$ define the local translation and rotation of joint $j$ at frame $f$ relative to its parent. In this work, we focus on a simplified setting that considers only the temporal evolution of rotations, $\mathbf{R}$, as dynamic features.

It is also important to note that working with skeletal rig representations of real-world 3D motions presents two key challenges. First, skeletal configurations vary significantly

due to anatomical differences across objects and the rigging styles of different experts, leading to variations in the number of joints and rest poses. Second, even identical motion dynamics can result in different joint rotations ($\mathbf{R}$) depending on the underlying static features. Thus, the ability to comprehend and generalize across diverse skeletal configurations is crucial for accurately modeling motion while preserving consistency across different structures.

## 4. Large-Vocab Text-to-Motion Dataset

Large-vocabulary object motion synthesis is challenging due to the diversity of objects and skeletal configurations. In this section, we discuss and address three key challenges for effective text-to-motion synthesis. First, it necessitates high-quality motion data that captures a variety of objects and motion patterns. Second, this motion data must be represented across diverse skeletal templates to account for real-world scenarios. Third, rich annotations with detailed text descriptions are essential to ensure that synthesized motions align accurately with the conditioned input text.

### 4.1. High-quality Motion Data

To address the first challenge, we utilize the Truebones Zoo dataset (Truebones, 2022), which contains over 1,000 artist-created animated armature meshes in FBX format, as illustrated in Figure 1. The dataset spans 70 unique animal species, including mammals, reptiles, birds, fishes, insects, and dinosaurs. Each species is uniquely characterized by its skeletal topology and rest pose, capturing the diversity of anatomical structures. Also, each animal object is associated

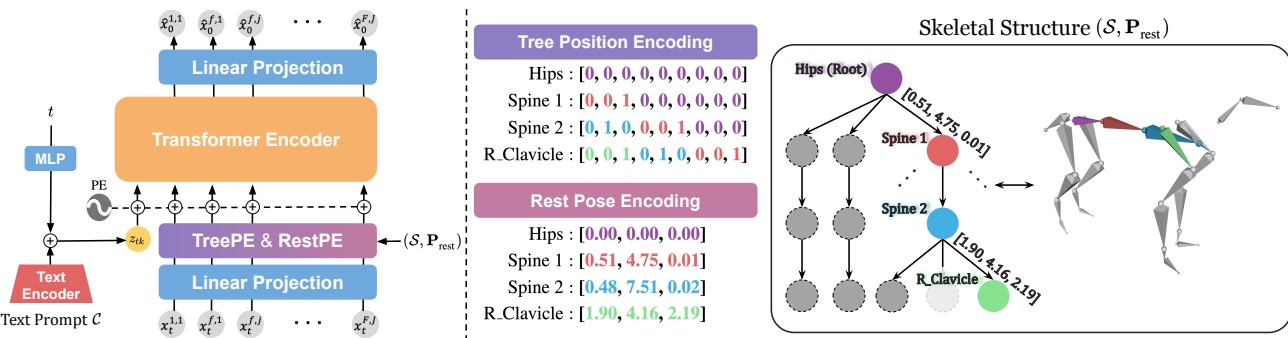

Figure 3. **Generalized Motion Diffusion Model**. **Left**: Our model conditions on skeletal information ($\mathcal{S}$, $\mathbf{P}_{\text{rest}}$) to adapt to diverse skeletal templates and a textual prompt $\mathcal{C}$ for text-driven synthesis. At each denoising step, the textual prompt $\mathcal{C}$ is encoded using the SigLip (Zhai et al., 2023) text encoder and the denoising timestep $t$ is embedded to form a conditioning token $z_{tk}$ that guides motion synthesis. **Right**: The topological ($\mathcal{S}$) and rest pose information ($\mathbf{P}_{\text{rest}}$) are encoded by Tree Positional Encoding (TreePE) (Shiv & Quirk, 2019) and Rest Pose Encoding (RestPE) and added to the noisy latent motion $x_t^{(f,j)}$ for the $f$-th frame and $j$-th joint. The enhanced representations, along with the conditioning token, are then processed by transformer blocks to estimate clean motions $\{\hat{x}_0^{(f,j)}\}_{f,j}$.

with multiple animations, ranging from 3 to 70 sequences.

From these FBX files, we extract armature animation data, capturing both the static structure and dynamic features, which are then preprocessed to construct our dataset. For preprocessing, we standardize each skeleton's rest pose by aligning its forward direction to the Y-axis and the upward direction to the Z-axis, and normalizing its scale and center to fit within a unit-length cube centered at the origin.

### 4.2. Rig Augmentation

Moreover, we introduce three rig augmentation techniques to simulate various skeletal rig configurations found in real-world motion data, as illustrated in Figure 2. These techniques, Bone Length Augmentation, Joint Number Augmentation, and Rest Pose Resetting, expand the diversity of skeletal configurations in the Truebones Zoo dataset while preserving the original motion dynamics.

**Bone Length Augmentation** randomly adjusts joint-to-joint bone lengths to simulate anatomical variations. To ensure natural motion after the adjustments, we leverage GPT-4o to categorize joints into specific body parts based on hierarchical information, such as joint names and parent-child relationships. Furthermore, GPT-4o determines appropriate scaling ratios for each part, ensuring the adjusted skeleton resembles anatomically similar objects (e.g., adjusting a dog's bone lengths to match those of a wolf). Each part is then scaled accordingly, with proportional and symmetrical adjustments applied to left and right counterparts to maintain anatomical balance and realistic motion dynamics.

**Joint Number Augmentation** modifies the skeletal rig by altering its structural complexity and organization. To adjust skeletal granularity, joints are randomly removed or subdivided, creating simpler or more detailed rigs.

**Rest Pose Resetting** alters the default configuration of the skeletal rig by changing the root orientations or joint angles in the rest pose. To determine a new rest pose, we randomly select a frame from the motion sequence and set the pose at that frame as the new rest pose.

### 4.3. High-Quality Multi-Level Text Annotation

High-quality text annotations are essential for ensuring precise alignment between textual descriptions and motion data. For this, we manually annotate the motion data in the Truebones Zoo dataset with detailed, high-quality descriptions. These annotations are delivered in a multi-level format, providing varying degrees of detail to address different levels of abstraction and precision required by the model.

Examples of these multi-level descriptions are displayed in Figure 2. To elaborate, the captions are structured into three levels of detail, short, mid, and long, progressively increasing in the amount of information they provide about the motion. Short captions deliver only the high-level actions, while details on initial postures, and part-level or directional dynamics are gradually added in mid and long captions.

For interested readers, detailed statistics on the dataset are provided in the appendix. These include the full list of animal categories in Truebones Zoo dataset (Table 3), word count histograms for each description level (Figure 10), and statistics on the most frequent verbs and nouns used in the annotations (Figure 11 and Figure 12, respectively).

## 5. Generalized Motion Diffusion Model

We introduce a generalized motion diffusion model that extends single-object motion synthesis to diverse objects with arbitrary skeletal templates. Figure 3 provides an overview of our framework. Unlike prior works (Tevet et al., 2023; Raab et al., 2024), which assume a fixed skeleton topology, our key innovation is the explicit incorporation of skeletal

configuration information through Tree Positional Encoding (TreePE) and Rest Pose Encoding (RestPE), enabling motion generation across varying skeleton templates. We first present the background on single-object motion diffusion models before detailing our generalization extensions.

## 5.1. Motion Diffusion Models

Diffusion models (Song & Ermon, 2020; Ho et al., 2020) learn to generate realistic samples $x_0$ by progressively reversing a noising process that begins with random noise $x_T$. At each intermediate step $t$, the data is represented as $x_t$, gradually refined as the neural network $\epsilon_\theta$ denoises it step by step, which is trained to predict and remove the noise at each stage. To further control the denoising process with conditioning variables $\mathcal{C}$, classifier-free guidance (CFG) (Ho & Salimans, 2021) is often utilized.

In the motion synthesis literature, a fixed skeleton topology $\mathcal{S}$ with a constant rest pose $\mathbf{P}_{\text{rest}}$ is typically assumed, focusing solely on synthesizing the dynamic features of motion for a single object. The task is then simplified to generating the rotation sequence $x_0 = \mathbf{R} \in \mathbb{R}^{F \times J \times D}$ from noisy latent variables $x_T$, given an additional input condition $\mathcal{C}$.

Diffusion Transformers (DiTs) (Peebles & Xie, 2023) serve as the backbone for parameterizing the denoising network $\epsilon_\theta$ to capture spatiotemporal dependencies in motion data. Under the fixed skeleton assumption, the input $x_t \in \mathbb{R}^{F \times J \times D}$ is reduced via a linear projection (Tevet et al., 2023):

$$z_t = \text{LinearProj}(x_t) \in \mathbb{R}^{F \times D'}, \tag{1}$$

where the $J \times D$ joint dimensions are compacted into $D'$-dimensional features per temporal frame, while spatial dependencies among joints are captured during this projection. To capture the temporal dependency among frames, positional encodings (Vaswani et al., 2017) are then added:

$$\hat{z}_t^{(f)} = z_t^{(f)} + \text{PE}(f), \tag{2}$$

where $z_t^{(f)}$ is the nosiy latent token and $\text{PE}(f)$ encodes position features for frame $f$. The resulting representation $\hat{z}_t \in \mathbb{R}^{F \times D'}$ is processed by the series of transformer blocks and re-projected back to the original motion space:

$$\hat{z}_0 = \text{Transformers}(\hat{z}_t), \tag{3}$$
$$\hat{x}_0 = \epsilon_\theta(x_t, t, \mathcal{C}) = \text{LinearReproj}(\hat{z}_0) \in \mathbb{R}^{F \times J \times D}. \tag{4}$$

## 5.2. Extension to Arbitrary Skeleton Topology

Motion synthesis for diverse objects requires the ability to handle skeleton topologies $\mathcal{S}$ with varying number of joints and their dependencies. To do so, our framework is designed to preserve the joint dimension throughout the denoising process. Specifically, the linear projection layer in Eq. (1)

is modified to independently transform $D$-dimensional per-joint representations into $D'$-dimensional tokens:

$$z_t^{(f,j)} = \text{LinearProj}(x_t^{(f,j)}) \in \mathbb{R}^{D'}. \tag{5}$$

Here, $x_t^{(f,j)}$ represents the input for frame $f$ and joint $j$.

To further encode the topological dependencies among joints, we employ Tree Positional Encoding (TreePE) (Shiv & Quirk, 2019). For each joint, TreePE captures both absolute position and relative dependencies within the tree-like skeletal structure $\mathcal{S}$ by encoding its path from the root as a binary-like sequence of transformations that track parent-child relationships and preserve hierarchical information. To integrate this information, we apply a simple MLP layer to each joint's tree encoding, then add the resulting representation $\text{TreePE}(j) \in \mathbb{R}^{D'}$ to the corresponding token, ensuring the hierarchical structure is embedded into the model.

$$\hat{z}_t^{(f,j)} = z_t^{(f,j)} + \text{PE}(f) + \text{TreePE}(j). \tag{6}$$

## 5.3. Handling Arbitrary Rest Pose

In hierarchical skeletal rig system, the dynamic features of motions are determined not only by the skeletal topology but also by the specific rest pose of each object. To ensure accurate modeling of motion dynamics, our framework incorporates explicit encoding of rest pose information. Specifically, the rest pose offset $\mathbf{P}_{\text{rest}}(j) \in \mathbb{R}^3$ for joint $j$ is transformed into a positional embedding $\text{RestPE}(j) \in \mathbb{R}^{D'}$ by a simple MLP module with sinusoidal encoding (Mildenhall et al., 2020), capturing its relative position and geometry within the skeleton in 3D space. These embeddings are added to the corresponding joint tokens:

$$\hat{z}_t^{(f,j)} = z_t^{(f,j)} + \text{PE}(f) + \text{TreePE}(j) + \text{RestPE}(j). \tag{7}$$

Through this integration, our model explicitly captures the skeletal topology and its rest pose configuration, enabling accurate modeling of motion dynamics for diverse objects.

## 5.4. Two-Stage Learning Approach

While the proposed modifications enhance adaptability to diverse skeleton templates, they also increase computational costs due to the expanded joint dimension. To mitigate this, we draw inspiration from text-to-video diffusion models (Blattmann et al., 2023), adopting a two-stage learning approach with factorized spatial-temporal attention to decouple pose modeling from motion dynamics.

In the first stage, we train a pose diffusion model to learn joint rotation dependencies within each frame independently using spatial attention blocks. Multi-view rendered images of each pose are used as conditioning inputs (see Figure 9). Once trained, this model is frozen. In the second stage, we enhance the model by introducing temporal attention blocks

*Table 1.* Ablation study on the positional encodings (TreePE & RestPE) and the rig augmentation technique in pose-level diffusion models. R@1 represents top-1 R-Precision (retrieval accuracy), with the '+' symbol indicating evaluation on a dataset with rig augmentation applied. The first row represents the oracle performance, where retrieval and alignment are measured between paired ground-truth pose and image embeddings, reflecting the quality of the embedding space of the learned pose encoder.

| Ablated Components | | Train Set | | | Test Set | | | Test Set$^+$ | | |
| --- | --- | --- | --- | --- | --- | --- | --- | --- | --- | --- |
| PEs | Rig Aug. | FID ($\downarrow$) | R@1 ($\uparrow$) | Align. ($\uparrow$) | FID ($\downarrow$) | R@1 ($\uparrow$) | Align. ($\uparrow$) | FID ($\downarrow$) | R@1 ($\uparrow$) | Align. ($\uparrow$) |
| - | - | 0.0000 | 0.9751 | 0.9850 | 0.0000 | 0.8822 | 0.9151 | 0.0000 | 0.8734 | 0.9311 |
| ✗ | ✗ | 0.9203 | 0.3602 | 0.6755 | 2.2564 | 0.2626 | 0.6870 | 1.0188 | 0.3705 | 0.7422 |
| ✗ | ✓ | 0.9813 | 0.3322 | 0.6627 | 2.2714 | 0.2562 | 0.6763 | 0.8628 | 0.3867 | 0.7454 |
| ✓ | ✗ | **0.0070** | **0.9739** | **0.9797** | 0.7704 | 0.5979 | 0.8734 | 0.6871 | 0.4435 | 0.7706 |
| ✓ | ✓ | 0.0072 | 0.9693 | 0.9700 | **0.6802** | **0.6045** | **0.8890** | **0.2636** | **0.6707** | **0.9276** |

after each spatial attention block, allowing it to capture motion dynamics over time while leveraging the pose-level knowledge from the first stage. Additionally, we employ factorized spatial-temporal attention: spatial attention captures structural dependencies by enabling joints to attend to each other within a frame, while temporal attention operates on a per-joint basis to model motion trajectories across frames. We set $J = 150$ and $F = 90$ throughout the paper.

## 6. Experiments

All models were trained on a Linux system equipped with either an NVIDIA RTX A6000 (48GB) or A100 (40GB) GPU. The pose diffusion model required approximately 29GB of VRAM with a batch size of 512 over 400K iterations, completing training in roughly 30 hours. The motion diffusion model used about 38GB of VRAM with a batch size of 4 and sequence length of 90, trained for 1M iterations over approximately 4 days.

To ensure motion integrity of the augmented rigs, bone length adjustments are restricted to $0.8\times$–$1.2\times$ of the original and applied symmetrically when symmetry existed. Bone erasing was limited to distal appendages (toes, head tips, tail ends) and redundant spine/root bones. Under these constraints, we visually verified 10K augmented results and found them suitable for training.

### 6.1. Dataset & Evaluation Metrics

To evaluate the pose synthesis model, we aggregate all motion data for each object category and extract their poses. We then apply clustering, generating 30 distinct pose clusters per object. Three clusters are randomly selected as the test pose set, while the remaining clusters are used for training. For motion synthesis evaluation, we randomly select one motion per object for the test set, with the remaining motions used for training.

Next, we apply the data augmentation techniques introduced in Section 4.2 independently to each motion, expanding the dataset to include 2.5M poses and 26K motions, up from

the original 125K poses and 1K motions. Moreover, we use SigLIP-SO400M-patch14-384 (Zhai et al., 2023) to extract image and text embeddings to condition our pose and motion diffusion models, respectively. For further details on data construction pipeline, please refer to Appendix B.1.

We evaluate the models using six automated metrics, along with a user study, to comprehensively assess their performance. The metrics include Fréchet Inception Distance (FID) (Lee et al., 2019), R-Precision (Guo et al., 2022), Alignment (Guo et al., 2022), Coverage (Li et al., 2022), Multimodality (Lee et al., 2019), and Motion Stability Index (MSI) (Kim et al., 2024). FID measures the similarity between the feature distributions of generated and real data, with lower scores indicating more realistic outputs. R-Precision and Alignment evaluate how well generated samples match input conditions. R-Precision quantifies the proportion of correct matches, while Alignment measures cosine similarity, with higher values indicating better correspondence. Coverage assesses how much of the reference data is represented by the generated samples, considering a reference covered if its cosine similarity with generated data exceeds a threshold. Multimodality captures the diversity of generated outputs for a single input, with higher scores reflecting greater variability. MSI measures temporal smoothness and stability, with higher scores indicating less jitter and stable motion over time. Finally, we conduct a user study to evaluate perceptual quality. Given rendered motion videos from each method, 30 participants compare the outputs in terms of realism, alignment with the input prompts, and overall preference. We aggregate the results across participants and report the average preference scores.

For automated evaluation, we train and utilize a pose-level encoder to compute all metrics instead of a motion-level encoder due to the limited availability of motion data. To ensure the pose encoder captures nuanced and meaningful representations of poses, we align its embedding space with that of the pretrained SigLIP image encoder, enabling the pose encoder to inherit the rich semantic knowledge learned during large-scale pretraining. We refer the reader to Table 1

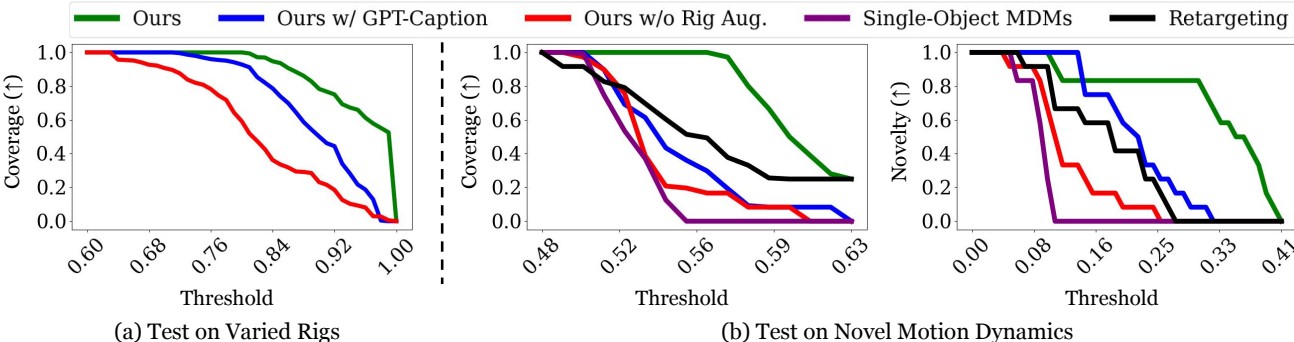

*Figure 4.* **Quantitative comparisons on text-to-motion synthesis scenarios.** A higher Area-Under-the-Curve (AUC) indicates the better adaptability to varied rigs or motion dynamics. Our approach enjoys greater generalization capability in synthesizing motions for diverse rigs (*i.e.* leftmost pane) and dynamics (*i.e.* middle and right panes) compared to the baselines.

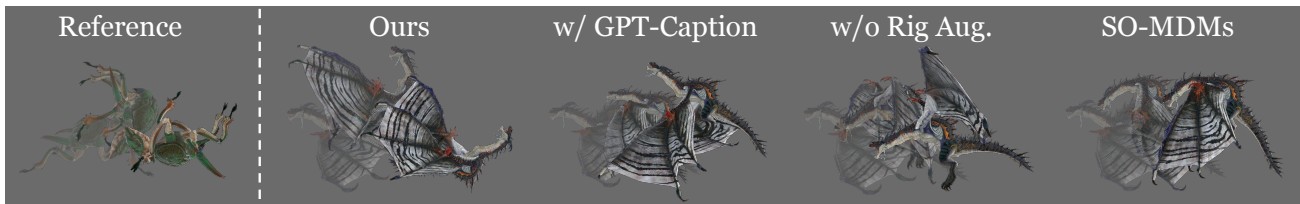

*Figure 5.* **Adaptability to novel motion descriptions.** We compare the synthesized motions of different models for a dragon using the novel motion description from the ant: "An ant is knocked back and ends up lying on its back, motionless." Among them, only our method successfully generates a dragon motion that closely aligns with the reference motion from the ant. Please watch the video in this link.

(*i.e.* 1st row) and Figure 13 for the quantitative and qualitative evaluation of the learned pose encoder, respectively. Also find Appendix B.2 for additional details regarding the architecture and training protocol.

## 6.2. Ablation Study

We first examine the necessity of the introduced positional encodings (PEs, *i.e.* TreePE and RestPE) and the rig augmentation technique in enhancing the adaptability of models to diverse object skeletons to improve overall performance. These experiments focus on the first-stage, pose-level diffusion models, with results presented in Table 1.

The results clearly demonstrate that models without PEs suffer from significantly degraded performance across all evaluation metrics and datasets. In particular, these models struggle to estimate rotations that produce poses well-aligned with the conditioned inputs, as reflected in low R-Precision (R@1) and Alignment scores. When combining both PEs and rig augmentation, the model achieves the best generalization performance. While PEs alone deliver substantial improvements by encoding essential rig-specific information, rig augmentation further enhances the model's robustness to diverse skeleton templates during training.

## 6.3. Text-to-Motion Synthesis

Next, we evaluate the models' ability to synthesize motions from text descriptions, focusing on two key generalization aspects: (1) adaptability to diverse rigs and (2) ability to syn-

thesize novel motions dynamics. As this is the first work on text-to-motion synthesis for large-vocabulary objects with varying skeletal templates, we compare our method against reasonable ablated baselines in this setup. The following sections provide further details.

**Adaptability to Varied Rig Styles.** To evaluate adaptability across diverse rig styles, we exclude one motion from the training set per object and augment its rig to create multiple test sets with consistent dynamics but varied configurations. Coverage is used as metric, measured as Area-Under-the-Curve (AUC) by sweeping the similarity threshold from 0 to 1. We compare our method with two ablated variants: one without rig augmentation and another using GPT-4o-generated captions. Results are presented in Figure 4-(a).

As shown in the figure, the model without the rig augmentation technique exhibits poor adaptability to diverse rig styles, indicated by the lowest AUC. Among others, the model trained with high-quality human-annotated descriptions achieves the largest AUC, validating the effectiveness of both the rig augmentation technique and the use of high-quality, human-annotated captions for enhancing adaptability and performance.

**Ability to Synthesize Novel Motions.** This evaluation tests the model's capacity to synthesize motion dynamics not included in the dataset for a particular object by leveraging textual descriptions from other objects. We introduce two additional methods: Single-Object Motion Diffusion Models (SO-MDMs) (Tevet et al., 2023), trained individually for

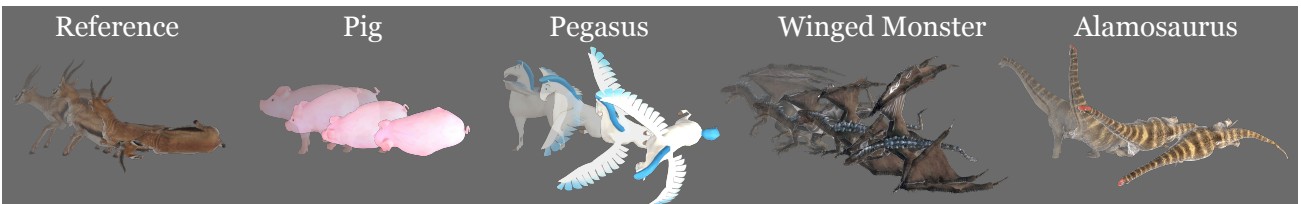

*Figure 6.* **Motion synthesis on novel objects and skeletons.** Surprisingly, our model generalizes well to objects with unseen skeletal structures and motion dynamics. Notably, while our dataset includes quadrupedal horses, bipedal reptiles, and dragons, it lacks examples that combine four legs with wings or feature winged reptiles. Additionally, no objects in the dataset possess necks as long as that of the Alamosaurus. All evaluation data were sourced from the web. For video demonstrations, please visit the project page in this link.

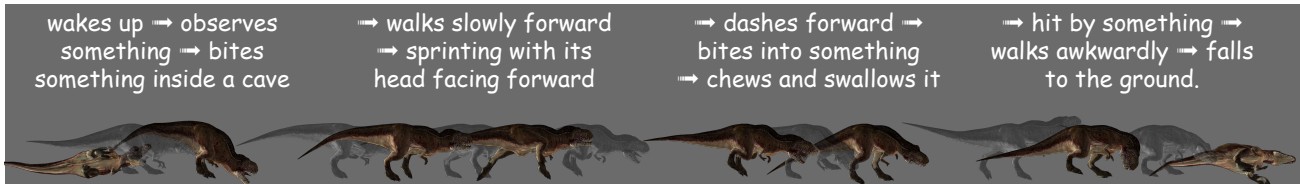

*Figure 7.* **Long motion synthesis.** Our model generates temporally coherent extended motions for story-level synthesis by conditioning on a sequence of descriptions and sampling overlapping short sequences, such as a T-rex's journey. Please watch the video in this link.

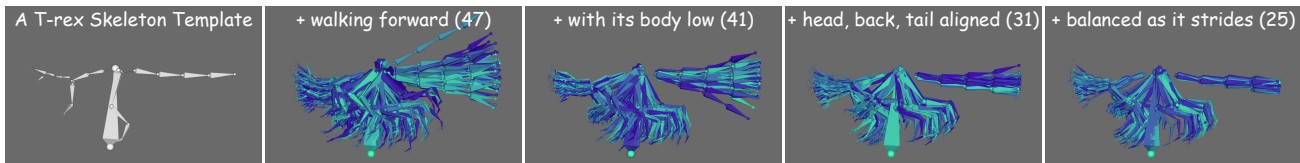

*Figure 8.* **Level of detail in description and variability in synthesized motions.** The value in parentheses represents the variability of motions synthesized from the same caption, measured using the multimodality metric with higher values indicate greater variability. The video can be viewed at this link.

specific objects, and a retargetting approach that bypasses learning entirely. Due to computational constraints, SO-MDMs are trained on five representative objects (Dragon, Tyrannosaurus, Cat, Cricket, and White Shark) chosen for their structural and dynamic diversity. We narrow the scope of this evaluation to these five objects accordingly.

Due to the limited scale of motion data, we do not exclude novel motion data from the training set for evaluation. Instead, we curate 10 key high-level actions that encompass the range of actions represented in the entire dataset. For each action, we randomly select two motion descriptions that align with the action from training objects not included in the five selected. These motions are treated as the target reference dynamics that the model should replicate when queried with the corresponding text descriptions.

We use coverage metric to assess how well generated motions align with reference dynamics. Additionally, we measure novelty by evaluating the inverse coverage of original motions of the chosen objects in the training set by their synthesized motions. Lower novelty scores indicate replication of training data, while higher scores reflect new, adaptive motions tailored to the queried descriptions.

Figure 4-(b) shows that our method achieves the highest AUC for both coverage and novelty, demonstrating supe-

rior alignment with queries while introducing novel motions. The analysis reveals two key insights: (1) The drop in performance for models using GPT-generated captions or trained without rig augmentation underscores the importance of high-quality human-annotated captions and rig augmentation, which help improving the model's understanding of motion dynamics and facilitating generalization across diverse rig styles. (2) The poor performance of SO-MDMs and retargetting approaches highlights the importance of amortized learning, which leverages shared structures across objects, enabling better generalization.

*Table 2.* User Preference and Smoothness (MSI) Scores.

| Method | Preference (%, ↑) | MSI ($\times 10^3$, ↑) |
|---|---|---|
| Ours | **65.60** | 8.78 |
| w/ GPT-Caption | 2.27 | 7.30 |
| w/o Rig Aug. | 12.40 | 7.39 |
| SO-MDMs | 10.67 | 7.21 |
| Retargeting | 9.07 | **9.93** |

Finally, we report comparison results from a user study and a motion smoothness evaluation (measured by MSI), summarized in Table 2. Our method is consistently preferred by participants in terms of alignment with text prompts and overall motion quality, receiving the highest preference

score by a substantial margin. Regarding smoothness, our approach achieves the highest score among all data-driven methods, indicating more temporally stable motions, while slightly trailing the retargeting method, which benefits from direct motion transfer but lacks flexibility to handle novel prompts. The results further highlight the effectiveness of our method in producing both perceptually preferable and plausible motions.

## 7. Additional Analysis

### 7.1. Generalization To Unseen Objects and Skeletons

We evaluate our model's ability to generalize to novel objects with entirely different species and rig configurations.

To this end, we obtain additional object meshes and armature data from external repositories such as SketchFab and use the curated text descriptions from Section 6.3 to generate motions. As shown in Figure 6, our model synthesizes coherent motions that align with reference dynamics, even for objects with previously unseen skeletal structures. This demonstrates its potential as a robust foundation for motion synthesis in open-vocabulary settings, expanding its applicability to diverse and novel object categories.

### 7.2. Long Motion Sequence Generation

Though trained on sequences up to $F = 90$ frames, our model extends to longer motions by conditioning on multiple sequential textual descriptions. To ensure smooth transitions, we apply weighted blending at the boundaries of consecutive motion chunks during sampling. As shown in Figure 7, our method enables the generation of extended, coherent motion sequences. This ability suggests promising applications in story-level motion synthesis (Zhou et al., 2024), where objects dynamically transition through complex, extended narratives, opening new possibilities for animation, virtual storytelling, and interactive experiences.

### 7.3. Generating Motions with Multi-Level Descriptions

We examine how textual detail impacts motion synthesis. Using a T-rex skeleton, we generate motions with progressively detailed descriptions. For each level, we sample 50 motions, measure variability using the Multimodality metric (higher values indicate greater diversity), and visualize their mid-frames in Figure 8. As shown, increased detail guides motion synthesis toward more structured and nuanced motion patterns, reducing ambiguity while preserving diversity. Thanks to the multi-level description training, our model effectively learns to generate both broad and fine-grained motions, demonstrating the importance of detailed and well-structured textual annotations in motion synthesis.

## 8. Discussion

We introduced a novel framework for text-driven motion synthesis across diverse object categories with heterogeneous skeletal structures. By augmenting the Truebones Zoo dataset with rich textual descriptions, introducing rig augmentation techniques, and extending motion diffusion models to dynamically adapt to arbitrary skeletal templates, our approach enables realistic, coherent motion generation for diverse objects. Experiments further demonstrate the potential of our method in generalizing to unseen objects, generating long motion sequences, and enabling controlled motion synthesis. We hope our work lays a strong foundation for motion synthesis across diverse object categories, advancing many applications in 3D content creation.

Despite its strengths, our method has a few limitations. First, for simplicity, our current implementation uses joint-wise Euler angle rotations only and ignores global root translation. However, this is not a fundamental limitation. Our model operates on motion tensors of shape $T \times J \times D$, where $D$ denotes joint features—in our case, rotations. This formulation can naturally be extended to include additional features such as global translation, relative translations for soft-constrained joints, or physically relevant signals like joint velocities or foot contact indicators. Including these signals may further improve temporal coherence, realism, and physical plausibility of synthesized motions.

Second, while our rig augmentation strategy enables generalization across diverse skeletal structures, the physical plausibility of augmented skeletons is limited. They may thus violate physical constraints (*e.g.* symmetry, balance, or anatomical consistency), limiting direct use in production-grade animation or simulation. Future work could improve plausibility via automated validation (*e.g.* foot-ground contact, joint velocity bounds, or end-effector stability), combined with rejection sampling or two-stage training. For example, pretraining on diverse augmented rigs for generalization, then fine-tuning on physically grounded data, may better balance flexibility and realism.

Third, while the method advances the field by offering a unified framework for motion synthesis across highly heterogeneous skeletal structures, it does not generalize well to human or open-vocabulary objects in zero-shot settings. This stems from the limited diversity, scale, and anatomical coverage of the training data: the Truebones Zoo dataset, though rich in non-human motions, lacks human-like skeletal structures and dynamics. Expanding to more balanced, large-scale motion corpora, such as Objaverse-XL (Deitke et al., 2024), could improve generalization to both human and novel object categories. This is a promising direction for future work, in line with recent advances in open-vocabulary dynamic mesh synthesis (Ren et al., 2023; 2024).

## Acknowledgement

Wonkwang Lee was supported by Institute of Information & communications Technology Planning & Evaluation (IITP) grant funded by the Korea government (MSIT) (No. RS-2022-II220156, Fundamental research on continual meta-learning for quality enhancement of casual videos and their 3D metaverse transformation). Wonkwang Lee thanks Chaewon Kim, Hakyoung Lee, Howon Lee, Hyeongjin Nam, JungSeok Cho, Kangwook Lee, Moonwon Yu, Suekyeong Nam, and Seongjoon Yang for their insightful feedback during the early stages of this work. He also thanks Dongmin Park, Gihyun Kwon, Hyuck Lee, Hyunjin Kim, Hyunseung Kim, Joo Young Choi, Juhyeong Seon, Jinseo Jeong, Minkyu Kim, and Sehun Lee for their helpful discussions during the development of this work.

## Impact Statement

This work advances the field of machine learning by introducing a novel approach for synthesizing motions across a broad spectrum of objects with diverse and arbitrary skeletal structures, enabling more flexible and generalizable motion generation. Potential applications include gaming, animation, virtual reality, and interactive experiences, broadening accessibility to motion synthesis tools.

While our method enhances creative applications, it also raises considerations regarding the ethical use of synthetic motion data, such as potential misuse in deceptive media or unintended biases in generated animations. We encourage responsible deployment and further research into bias mitigation and transparency in generative models.

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

# Appendix

This section presents additional details that are omitted from the main paper due to space constraints.

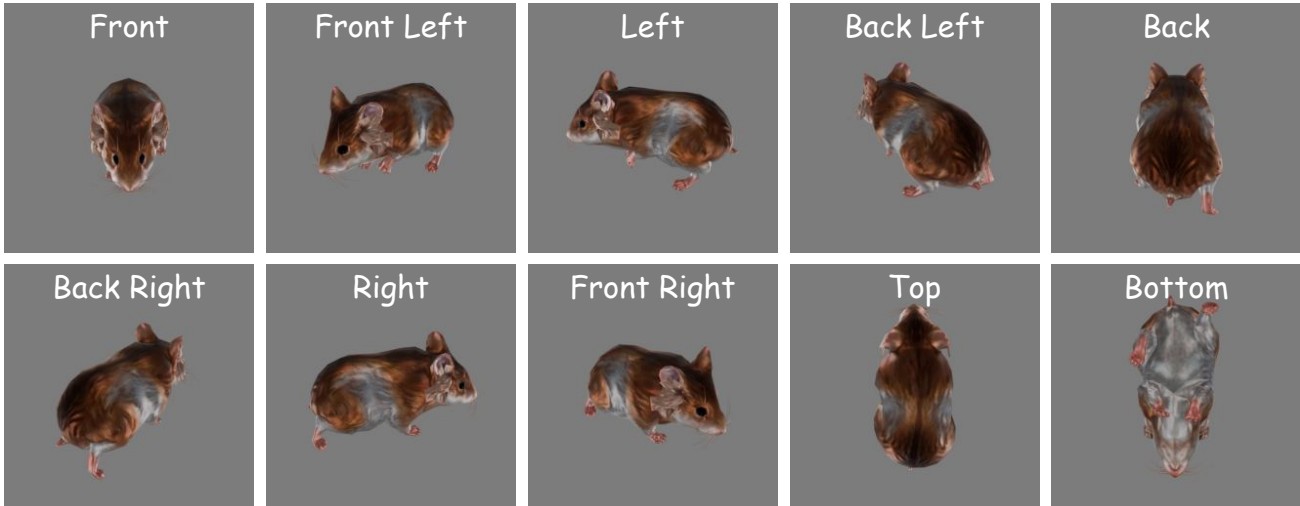

*Figure 9.* Rendered view images of a hamster's pose from ten predefined camera viewpoints. We use these rendering images to annotate descriptions for each motion and to train the pose encoder.

## A. Details on Data

**Source**    In this paper, we primarily utilize the Truebones Zoo (Truebones, 2022) dataset for training and evaluation. The types of objects and the number of 3D animations available for each object are summarized in Table 3. Each animation is provided in FBX format, consisting of an armature and its corresponding animation, with the armature rigged to a mesh.

**Preprocessing**    To prepare the data for training, we preprocess each sample by standardizing its orientation, scale, and center. Specifically, during preprocessing, we manually adjust the orientation of each motion to align the forward direction with the -Y-axis and the upward direction with the Z-axis. The armature and mesh are then uniformly scaled so that the longest dimension of their bounding box is normalized to 1. Finally, we reposition the object to be centered at the origin (0,0,0).

**Extracting Training Data**    After preprocessing, we extract the armature animation into BVH format, discarding joint positions and retaining only the bone hierarchy information (*i.e.* skeletal structures and rest poses) and joint rotations. These components are utilized as training features: the bone hierarchy information serves as static features, while joint rotations are treated as dynamic features. The joint rotations are represented using Euler angles in a ZXY order. Additionally, individual poses in each motion are rendered from ten distinct camera perspectives: front, front left, left, back left, back, back right, right, front right, top, and bottom. Please find Figure 9 for the example for a single pose. These rendered images are used to create motion description annotations and to train image-to-pose diffusion models.

**Annotation Pipeline**    We manually annotate high-quality motion descriptions for each motion sample. Human annotators are initially provided with rendered videos captured from the front-left camera view but are given access to additional views when the front-left perspective alone is insufficient to accurately identify the animation. Annotators are instructed to describe the motions in as much detail as possible, focusing on the following key aspects: (1) Initial Pose: a clear description of the starting pose of the object (*e.g.* initially standing), (2) High-Level Actions: the overarching actions being performed over time (*e.g.* standing, striking, etc.), and (3) Part-Level and Directional Dynamics: finer-grained details of motions, such as specific body part movements and directional actions (*e.g.* "Initially standing on all fours, an object is standing on hind limbs by pushing off the ground with its two front limbs, followed by striking by lifting its right front paw over its head").

Subsequently, these annotated captions are refined using GPT-4o to create multi-level descriptions. Specifically, the annotated captions are first revised to enhance clarity and ensure consistency across descriptions; these revisions are used as

*Table 3.* The types of objects, the number of the 3D animations, and thier number of joints in Truebones Zoo dataset (Truebones, 2022).

| Object Name | # Animations | Average # Joints | Object Name | # Animations | Average # Joints |
|---|---|---|---|---|---|
| Alligator | 21 | 21.00 | Jaguar | 14 | 40.00 |
| Anaconda | 18 | 27.00 | Great White Shark | 10 | 16.00 |
| Ant | 16 | 34.00 | King Cobra | 11 | 17.18 |
| Bat | 11 | 43.00 | Leopard | 14 | 43.00 |
| Bear | 30 | 62.00 | Lion | 15 | 26.00 |
| Bird | 19 | 53.00 | Lynx | 14 | 33.00 |
| Brown Bear | 22 | 32.00 | Mammoth | 12 | 36.00 |
| Buffalo | 18 | 36.00 | Monkey | 15 | 70.00 |
| Buzzard | 11 | 50.00 | Ostrich | 10 | 45.00 |
| Camel | 19 | 46.00 | Parrot-1 | 14 | 56.00 |
| Cat | 4 | 38.00 | Parrot-2 | 3 | 50.00 |
| Centipede | 9 | 64.00 | Pigeon | 12 | 7.08 |
| Chicken | 4 | 30.00 | Piranha | 11 | 22.00 |
| Komodo Dragon | 9 | 50.00 | Polar Bear | 9 | 36.00 |
| Coyote | 10 | 36.00 | Baby Polar Bear | 14 | 34.00 |
| Crab | 11 | 44.00 | Pteranodon | 13 | 34.00 |
| Cricket | 11 | 43.00 | Puppy | 4 | 34.00 |
| Crocodile | 12 | 34.00 | Reindeer | 9 | 32.00 |
| Crow | 10 | 23.00 | Velociraptor-1 | 10 | 33.00 |
| Deer | 21 | 37.00 | Velociraptor-2 | 39 | 49.00 |
| Dog-1 | 42 | 46.00 | Velociraptor-3 | 14 | 50.00 |
| Dog-2 | 36 | 46.00 | Rat | 7 | 13.00 |
| Dragon | 9 | 111.00 | Rhino | 9 | 38.00 |
| Eagle | 12 | 39.00 | Roach | 11 | 35.00 |
| Elephant | 13 | 32.00 | Sabre-Toothed Tiger | 45 | 54.00 |
| Fire Ant | 25 | 35.00 | Sand Mouse | 13 | 33.00 |
| Flamingo | 4 | 34.00 | Scorpion-1 | 13 | 53.00 |
| Fox | 13 | 35.00 | Scorpion-2 | 40 | 46.00 |
| Gazelle | 18 | 35.00 | Skunk | 8 | 29.00 |
| Giant Bee | 9 | 39.00 | Spider-1 | 33 | 57.00 |
| Goat | 11 | 27.00 | Spider-2 | 18 | 51.00 |
| Hamster | 5 | 38.00 | Stegosaurus | 10 | 35.00 |
| Hermit Crab | 12 | 55.00 | Tyrannosaurus Rex | 70 | 49.00 |
| Hippopotamus | 10 | 36.00 | Triceratops | 9 | 27.00 |
| Horse | 29 | 63.00 | Toucan | 9 | 15.00 |
| Hound | 12 | 40.00 | Turtle | 10 | 39.00 |
| Termite | 9 | 52.00 | Tyrannosaurus | 10 | 54.00 |

the long descriptions. Next, GPT-4o is tasked with producing coarser-level descriptions by progressively omitting part-level or directional details from the long descriptions, resulting in mid-level and short descriptions.

For a visualization of word count distributions across description levels, see Figure 10. Statistics on the most frequent verbs and nouns are presented in Figure 11 and Figure 12, respectively.

**External Test Data**  We downloaded and used four armatured object meshes from Sketchfab: Pig (Harris, 2021), Icy Dragon (chengzijieczj, 2020), Pegasus (Batuhan13, 2021), and Alamosaurus (robertfabiani, 2018) to evaluate our model's ability to generate motions for novel objects.

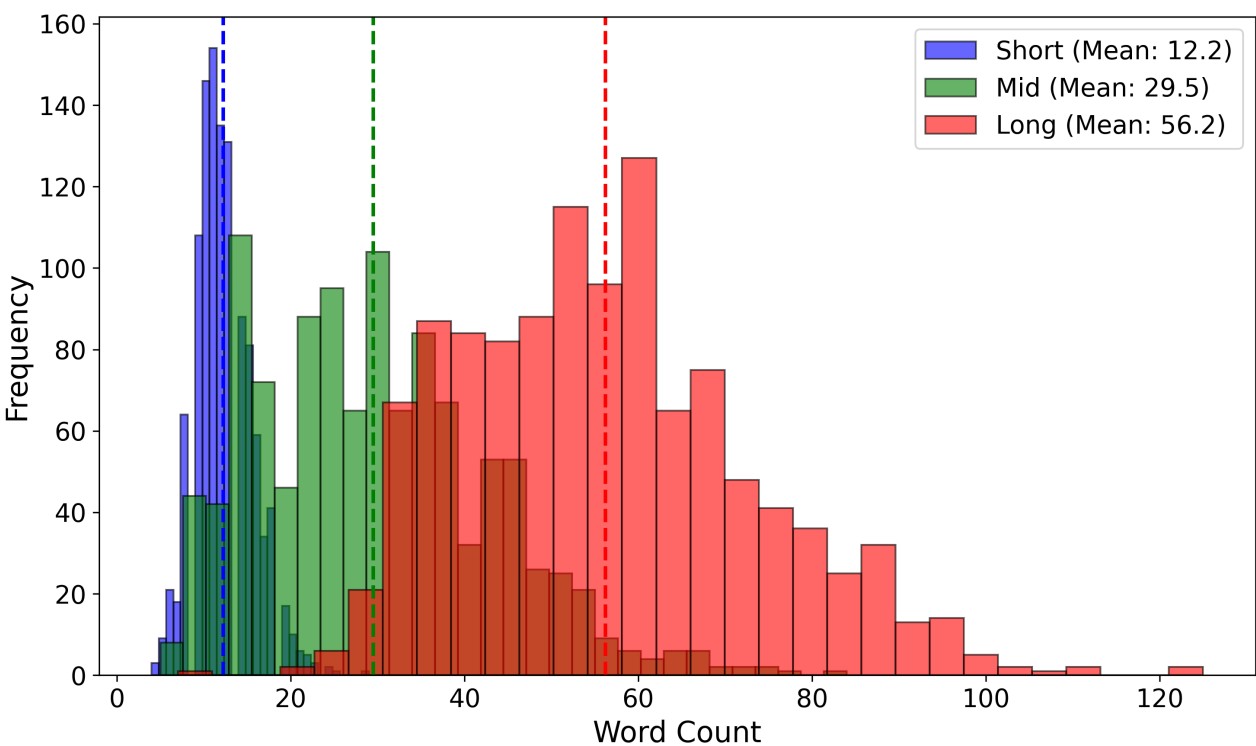

*Figure 10.* Word count histogram by description type.

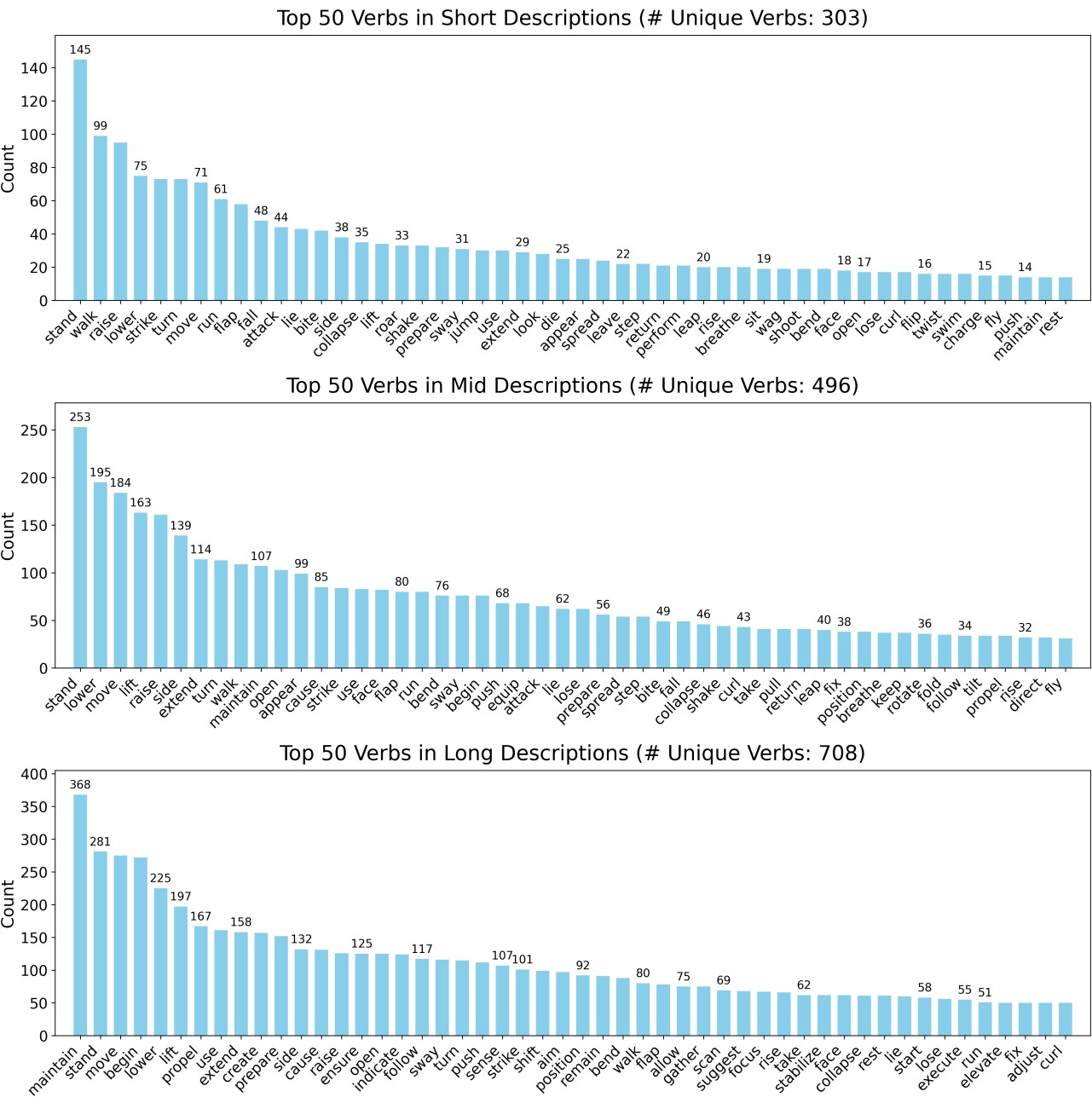

*Figure 11.* Top 50 verbs and their counts in our annotated motion descriptions.

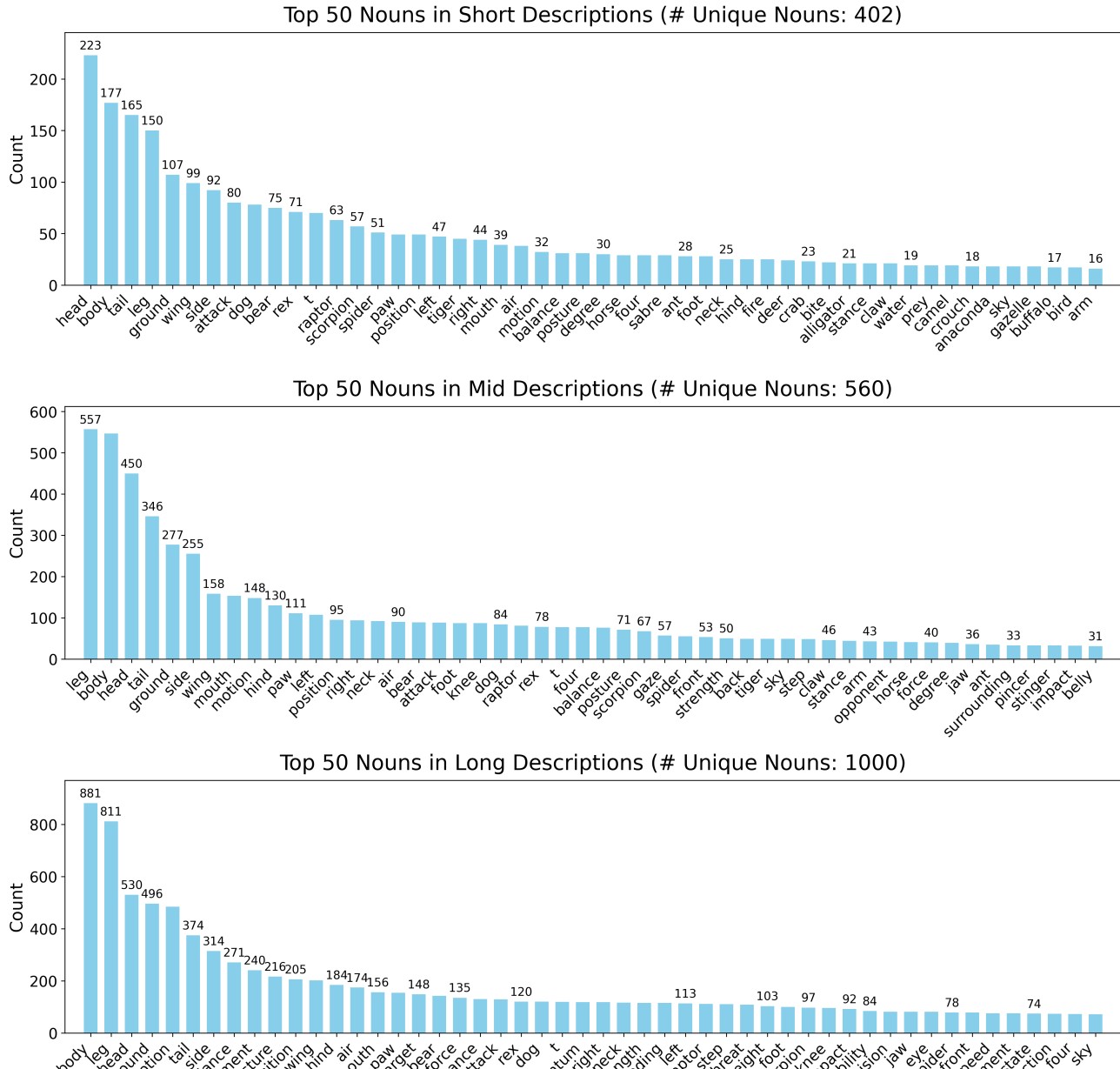

*Figure 12.* Top 50 nouns and their counts in our annotated motion descriptions.

# B. Details on Implementation & Training

## B.1. Generalized Motion Diffusion Models (G-MDMs)

**Architecture**  We adapt and modify the SiT-S (Ma et al., 2024) model to align with our setup, configuring the depth, hidden dimension, and the number of self-attention heads to 12, 384, and 6, respectively. On top of this, we introduce additional encoding layers to enhance the model's functionality: the condition encoding layer, tree position encoding (TreePE) layer, rest pose offset encoding (RestPE) layer, and frame position encoding layer. Each layer utilizes a Linear-SiLU-Linear block to project input features into a $D$-dimensional hidden space. Specifically, the condition encoding layer maps features from either the image or text encoder of the SigLIP-SO400M-patch14-384 (Zhai et al., 2023) model into the diffusion model's hidden dimension. TreePE and RestPE encode spatial positional information, while the frame position encoding layer captures the temporal sequence of motion.

To handle the variable number of joints and frames in motion data, we set the maximum number of joints and frames to 140 and 90, respectively. The maximum joint count reflects the largest number of joints observed in the dataset, while the maximum frame count was manually set to 90 after observing the motion sequences as a reasonable cutoff. During training, inputs with fewer joints or frames than these maximum values are padded with zeros, and the padded elements are masked within the self-attention modules to ensure they do not influence the diffusion process. For motions containing more than 90 frames, we randomly sample a chunk of frames during training. To condition the model on rendering images or text, we adopt the adaLN-Zero conditioning approach (Peebles & Xie, 2023), which has demonstrated strong effectiveness in integrating conditional embeddings into the model.

**Two-Stage Training**  To address the challenges posed by limited computational resources and potential overfitting due to the small size of the training motion dataset (approximately 1K samples), we draw inspiration from text-to-video diffusion models (Blattmann et al., 2023; Wang et al., 2023) and adopt a two-stage training approach alongside factorized attention mechanisms. This approach decouples the modeling of poses from motion dynamics, effectively reducing computational complexity while maintaining adaptability and generalization.

In the first stage, we train a pose diffusion model to capture dependencies among joint rotations within each frame using spatial attention blocks. Instead of relying on textual descriptions, the pose synthesis model is conditioned on average embeddings of poses image renderings viewed from ten predefined camera angles (*e.g.* front, front left, left, back left, back, back right, right, front right, top, bottom). Once the pose diffusion model is trained, it is frozen.

In the second stage, temporal attention blocks are introduced after each spatial attention block. These temporal attention blocks enable the model to capture motion dynamics across time, leveraging the pose-level knowledge acquired during the first stage. This staged training approach reduces computational overhead while ensuring the synthesis of coherent and realistic motion sequences.

To further manage computational complexity, we adopt a factorized attention mechanism inspired by text-to-video diffusion models. In this setup, spatial attention operates within each frame, allowing joints to attend to one another and effectively capturing structural dependencies. Temporal attention, on the other hand, operates across frames, enabling each joint to focus on its motion trajectory over time. By separating spatial and temporal attention, the model reduces the computational cost of full spatiotemporal modeling while maintaining the capacity to generate realistic and temporally coherent motions.

## B.2. Pose Encoder

We train an additional pose-level encoder network to facilitate the evaluation of the pose and motion diffusion models. The encoder adopts the same architecture as our pose-level generalized diffusion model, as detailed in Section B.1.

The forward pass of the pose encoder is identical to that of the pose diffusion model, except that the pose encoder's output is derived by average pooling the pose representations along the joint dimension. Specifically, given two types of inputs—(1) static feature information, represented by TreePE and RestPE, and (2) dynamic feature information, either real or synthesized, represented as rotation values with a shape of $(B \times J \times 3)$—the pose encoder pools the output along the joint dimension to produce a final representation of shape $(B \times D)$. This representation is then projected into the feature space of SigLIP-SO400M-patch14-384, serving as pose embeddings.

The pose encoder is trained to align its embeddings with those of the image encoder of SigLIP-SO400M-patch14-384 using contrastive learning. To improve this alignment, we introduce a learned view image embedding layer. Specifically,

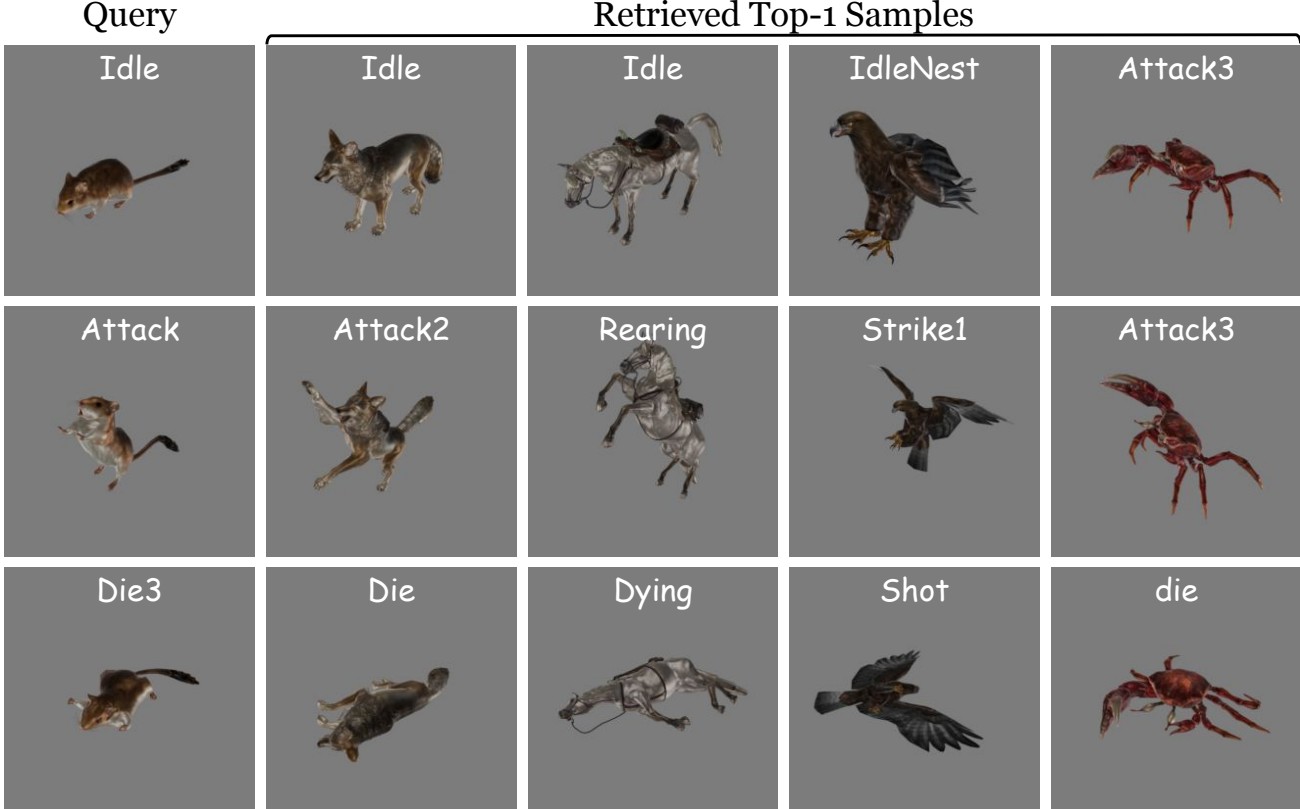

*Figure 13.* Qualitative evaluation of the learned pose embedding space. Given a query pose of a sandmouse, we present rendered images of the top-1 retrieved poses from four distinct objects: coyote, horse, eagle, and crab. We display the file name of the motion file containing each retrieved pose. These results highlight the ability of the learned pose encoder to effectively embed similar poses from different animal objects into the same embedding space, capturing meaningful semantic relationships within the pose space.

for each pose, a paired set of ten camera views, as described in Section A, is embedded into image embeddings using the pretrained image encoder. Camera position embeddings are then added to these view embeddings, followed by two layers of self-attention blocks. The outputs are average pooled to produce a unified image embedding, which serves as the target for alignment with the pose embeddings.

To evaluate the quality of the learned pose embedding space, we perform both quantitative and qualitative analyses. For the quantitative evaluation, which measures the alignment performance of the trained pose encoder, please refer to Table 1. The qualitative results can be found in Figure 13.

As evident from both evaluations, our pose encoder demonstrates strong alignment with the aggregated image embeddings of the pretrained image encoder (e.g., high retrieval and alignment scores in the table). Furthermore, it has learned meaningful semantic structures within the pose embedding space, effectively identifying and grouping similar poses of different objects within this space.

## C. Details on Coverage and Novelty Metrics

In human motion synthesis, it is common practice to train and utilize a motion encoder that projects motion sequences into an embedding space, where evaluation metrics are subsequently computed. However, due to the limited availability of motion data in our setup, we instead leverage a learned pose encoder to assess the quality of the synthesized motions. We adapt the original coverage metric (Li et al., 2022) to our setup, making slight modifications to better suit our motion synthesis evaluation.

Given a reference motion $x$ and a synthesized motion $\hat{x}$ of lengths $L_x$ and $L_{\hat{x}}$, respectively, we extract all possible windows of size $F_w$ from each motion:

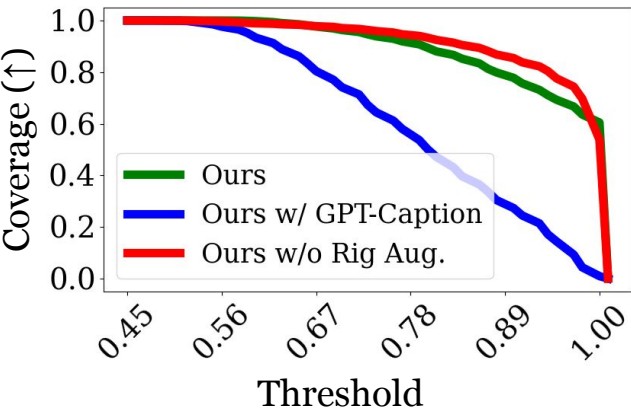

Figure 14. Quantitative comparisons of the models on the training set, evaluated using the coverage metric. While our model achieves comparable performance in fitting the training data compared to the version without rig augmentation, our full approach demonstrates significant gains in generalization (*i.e.* Figure 4).

$$\mathcal{W}(x, F_w) = \{x^{i:i+F_w-1}\}_{i=1}^{L_x - F_w + 1}, \quad \mathcal{W}(\hat{x}, F_w) = \{\hat{x}^{i:i+F_w-1}\}_{i=1}^{L_{\hat{x}} - F_w + 1}. \tag{8}$$

Here, the window size $F_w$ is determined as $F_w = \min(L_x, L_{\hat{x}}, F)$, where $F = 90$ is the predefined maximum frame length for our generalized motion diffusion model.

**Coverage**  To assess how well the synthesized motion $\hat{x}$ covers the reference motion $x$, we compute the coverage metric as:

$$\mathrm{Cov}(\hat{x}, x; \Theta) = \frac{1}{|\mathcal{W}(x, F_w)|} \sum_{x_w \in \mathcal{W}(x, F_w)} \mathbb{1}\left[ \max_{\hat{x}_w \in \mathcal{W}(\hat{x}, F_w)} \mathrm{CosineSimilarity}(x_w, \hat{x}_w) > \Theta \right]. \tag{9}$$

This metric quantifies the proportion of reference motion windows $x_w$ that have at least one synthesized counterpart $\hat{x}_w$ with a cosine similarity above a threshold $\Theta$. To obtain a comprehensive evaluation, we sweep $\Theta$ from 0 to 1 and compute the area under the curve (AUC) of the coverage function, providing an aggregate measure of how well the synthesized motion spans the reference motion across different similarity thresholds.

**Novelty**  We also introduce the novelty metric, which measures how distinct the synthesized motion $\hat{x}$ is from the reference motion $x$. Higher novelty values indicate that the generated motions introduce new patterns rather than replicating the reference.

$$\mathrm{Nov}(\hat{x}, x; \Theta) = \frac{1}{|\mathcal{W}(\hat{x}, F_w)|} \sum_{\hat{x}_w \in \mathcal{W}(\hat{x}, F_w)} \mathbb{1}\left[ (1 - \max_{x_w \in \mathcal{W}(x, F_w)} \mathrm{CosineSimilarity}(x_w, \hat{x}_w)) > \Theta \right]. \tag{10}$$

This metric captures the proportion of synthesized motion windows $\hat{x}_w$ that do not closely resemble any reference window $x_w$, ensuring that the generated motions exhibit novelty. As with coverage, we sweep $\Theta$ from 0 to 1 and compute the AUC of the novelty function to obtain an overall measure of the diversity of the synthesized motion.

