# OpenReview forum: "How to Move Your Dragon: Text-to-Motion Synthesis for Large-Vocabulary Objects"
_ICML.cc/2025/Conference — ICML 2025 poster_

### Official Review · Reviewer_qPq2 · 2025-03-12

**Overall Recommendation:** 4

**Summary:**

The paper aims to generate motions of different rigs from text inputs. The paper witness the problem in the current 3D content creation community, that it lacks high-quality motion dataset with annotations, and it lacks methods handling hetergeneous skeleton templates. Therefore, the paper present a high-quality text-annotated text-to-object motion dataset, and a motion diffusion model that supports different skeleton templates. Specifically, the paper repsents three rig augmentation methods, and present incorporating Tree Position Encoding to extend MDM incoporating different skeleton structures. Experiment comparing with several component-removed baselines and motion retargetting, proving the effectiveness of the rig augmentation strategy, and generalized motion diffusion models. Experiment with partial data also presents the adapbility to various rigs, motions, and unseen objects.

**Claims And Evidence:**

The claims are well supported.

**Essential References Not Discussed:**

Please include these works that has the ability to generate character motion with different rigs, or overlap with the rigging augmentation techniques proposed in the paper.
(1) CharacterMixer: Rig-Aware Interpolation of 3D Characters
(2) SkinMixer: Blending 3D Animated Models

Please also include some paper using surface-representation or multi-view image representation for text to motion synthesis. Here I name some as an example:
(1) L4GM: Large 4D Gaussian Reconstruction Model
(2) DreamGaussian4D: Generative 4D Gaussian Splatting

**Experimental Designs Or Analyses:**

+ The paper compares the baseline with other methods using 'pose-level' encoders but not 'motion-levels'. Though I acknowledge that it's hard to train such a motion-level encoder, but the paper needs to provide evaluation in terms of the quality of motion, such as motion smoothness, or even perceptual study.

**Methods And Evaluation Criteria:**

+ My major concern here is on the physical playsibility of the augmented rigs. As these rigs are augmented by language models, it could potentially breaks the physical plausibility of the original rigs (i.e. uneven lengths feet leads to unstable standing, bones might intersect). Does the paper provide evaluation for these properties? I acknowledge that even though without physical playsibility, a MDM can still benefit from the augmented skeletons. However, are these augmented rigs really animation-usable?

**Other Comments Or Suggestions:**

+ The paper using the phrase "large-vocab" to describe the text-to-motion dataset. However, detailed stats of word frequency and word count are not provided.
+ Please specify the RestPE in the supplementary material, if it is not provided.

**Other Strengths And Weaknesses:**

+ I can see the paper, especially the rigging augmentation part, might also have broader impact in articulated object augmentation, that might benefit 3D design or robotics community.
+ The paper would also be interesting in studying skill transfer across embodiments, which could might interested robotics community.

**Questions For Authors:**

+ Are examples in figure 1 generated results or examples from the dataset? Please specify.
+ The dataset uniformed all skeletons as mentioned in section 4.1. However, since different creatures has different sizes, will uniforming object sizes confuses GPT or any text-related components, i.e., text-to-motion model?

**Relation To Broader Scientific Literature:**

+ The dataset would be the major contribution, which might interesting both animation and robotics community.
+ The cross-skeleton MDM model is also interesting.

**Theoretical Claims:**

+ Since different objects might behaves drastically different with the same verb, (i.e. how scorpion and horse attack looks drastically different as shown in Figure 10), will these actually harms the cross embodiement generalizability of the MDM?

---

> ### Author Rebuttal · Authors · 2025-03-31
>
> We deeply appreciate the reviewer’s detailed review and thoughtful insights.
>
> ### Q1: Physical plausibility of the augmented rigs.
> As noted in our response to Q5 of Reviewer 2SeY, we carefully designed the augmentation pipeline and visually inspected ~10K augmented motions, which we found sufficient at this scale. The resulting skeletons appeared visually plausible and were successfully used for training.
>
> That said, we acknowledge that these rigs were not derived from physically simulated or animation-ready meshes, and may not be suitable for production use. As discussed in response to Q3 of Reviewer 4SvT, their primary goal was to increase skeletal diversity and improve model generalization—e.g., robustness to unseen rigs or cross-object transfer.
>
> ### Q2: Does using the same verb for semantically different motions across objects hurt cross-embodiment generalizability?
> Rather than harming generalization, we believe this diversity encourages the model to learn embodiment-aware interpretations of similar textual prompts.
>
> The key lies in context-awareness. While both model design and data quality are important, we view this primarily as a data-centric challenge—facilitated by the adoption of a proven architecture (e.g., DiTs)—and best addressed through exposure to diverse, high-quality, and context-rich examples.
>
> To this end, we provide multi-level captions that describe actions, part-level dynamics, and initial poses. As shown in Section 7.3, these annotations help the model learn contextualized verb semantics and produce more structured motions.
>
> As a promising direction, scaling up with diverse resources like Objaverse-XL (~100K animated 3D meshes), along with automated pipelines for multi-level captioning, could greatly enhance cross-embodiment generalization through fine-grained contextual understanding. We opt to explore this as our future work.
>
> ### Q3: Additional evaluation in terms of the quality of motion.
> As suggested, we conducted additional evaluations on the test sets from Figure 4-(b), where all baselines are available.
>
> For motion smoothness, we used the Motion Stability Index (MSI) [1]; higher values indicate smoother, more stable motion. GT scored 12.01e3. Results:
> - Retargeting: 9.93e3, Ours: 8.78e3, NoAug: 7.39e3, GPT-Caption: 7.30e3, SO-MDMs: 7.21e3.
>
> Our method achieves the highest MSI among data-driven baselines, closest to GT. Retargeting scores highest overall due to direct motion transfer but lacks flexibility for novel prompts.
>
> We also conducted a user study with 30 participants, who selected the most caption-aligned motion. Results:
> - Ours: 65.6%, NoAug: 12.4%, SO-MDMs: 10.667%, Retargeting: 9.067%, GPT-Caption: 2.267%.
>
> These results show that our method produces smooth and semantically accurate motions, both quantitatively and perceptually. We will include them in the revision.
>
> References:
>
> [1] Kim et al., Audio-driven Talking Face Generation with Stabilized Synchronization Loss, CVPR 2023.
>
> ### Q4: Essential references not discussed & details on RestPE / Figure 1.
> We will include the relevant references in the revised manuscript. Details on RestPE will be added to the supplementary material. Also, the examples in Figure 1 are from the dataset (not generated); we will clarify this to avoid confusion.
>
> ### Q5: Word stats to support “large-vocab” claim.
> In our work, “large-vocab” refers to the diversity of object categories and skeletal structures, 70+ distinct categories with unique topologies, rather than textual or action diversity within a fixed skeleton, as in prior human motion literature.
>
> That said, we appreciate the suggestion and will include text-level stats in the revision. As a preview:
> - Avg. words per caption: short (12.2), mid (29.5), long (56.2)
> - Verb counts: short (302), mid (495), long (707)
> - Noun counts: short (402), mid (560), long (1000)
> - Top verbs (short): stand (145), walk (99), raise (95), lower (75), strike (73)
> - Top nouns (short): head (223), body (177), tail (165), leg (150), ground (107)
>
> For reference, the full list of object categories is also provided in Appendix Table 2, which may help clarify the intended meaning of “large-vocab” in our current draft.
>
> ### Q6: About normalizing scales of different creatures.
> We apply size normalization, motivated by early observations that it led to faster loss convergence during training.
>
> Nonetheless, we believe normalization does not hinder the model’s ability to learn size-specific motion patterns when such cues are present in the data. For example, even if a small and large cat are normalized to the same scale, their motions may still differ in agility, stride, or posture—captured through joint rotations, or temporal dynamics and timing. Although our annotated captions do not specify size explicitly, skeletal topology and relative proportions are preserved, providing structural cues that help the model infer such differences.

---

### Official Review · Reviewer_2SeY · 2025-03-12

**Overall Recommendation:** 3

**Summary:**

This paper proposes a novel problem, text-driven motion synthesis of different skeletal structures, constructs a dataset, and develops a new model structure. The key innovation is the explicit incorporation of skeletal configuration information through Tree Positional Encoding (TreePE) and Rest Pose Encoding (RestPE). Experiments show the  method generates realistic, coherent motions from textual descriptions for diverse and even unseen objects, setting a strong foundation for motion generation across diverse object categories with heterogeneous skeletal templates

**Claims And Evidence:**

Overall, the claims are almost supported. However, I still suspect the generalizations. Could the author provide some human motion in a zero-shot way?

**Essential References Not Discussed:**

No problem here.

**Experimental Designs Or Analyses:**

Why did the author choose PE to encode the rest pose? Does the author try other ways, like attention?

**Methods And Evaluation Criteria:**

The methods are good. And some human user studies may needed for evaluation.

**Other Comments Or Suggestions:**

Typo: 1. Page 2, Line 93, 'Similarly' should not be a reference.

**Other Strengths And Weaknesses:**

Strengths: This paper is interesting and meaningful. The dataset and the architecture design make sense.
Weaknesses: See the questions.

**Questions For Authors:**

1. Could the author provide the video demos link again? I could not open the link in the main paper PDF file.
2. Can this model generate human skeleton motions in a zero-shot way?
3. I did not get the process of rig augmentation. Did the author do retargetting after the augmentation? If yes, how do you keep the motion quality here? If not, could you explain further how to adjust the original motion to the new skeleton?
4. Could the author separate the TreePE and TestPE ablations to see what actually works?

**Relation To Broader Scientific Literature:**

The previous method makes it hard to process different skeletons in one model. This paper proposes to using PE to solve it.

**Theoretical Claims:**

No problem here.

---

> ### Author Rebuttal · Authors · 2025-03-31
>
> We sincerely thank the reviewer for the valuable feedback and questions.
>
> ### Q1: Typo & Link for the demo.
> We will correct the typo. For the demo, please visit: t2m4lvo.github.io
>
> ### Q2: Human study.
> Please refer to our response to Q2 of Reviewer qPq2, where we address a similar point.
>
> ### Q3: Why choose to use PE to encode the rest pose.
> We opted not to explore alternative methods, as our primary goal was to preserve the scalability and architectural simplicity of the backbone, Diffusion Transformers (DiTs/SiTs), which are known for their scalable and generalizable design. Positional Encoding (PE) offered a straightforward and effective way to incorporate rest pose information while maintaining full compatibility with our transformer-based framework. That said, we acknowledge that exploring alternative conditioning strategies could be a fruitful direction for future work.
>
> ### Q4: Zero-shot inference on human
> While our method is designed to accommodate diverse and heterogeneous skeletal templates, we find that it does not extend to human skeletons. This limitation stems from the characteristics of our training data: the Truebones Zoo dataset, even with rig augmentation, does not contain human-like skeletal structures. Additionally, the motion dynamics and textual descriptions in our dataset differ significantly from those in human motion domains.
>
> A natural question is why human motion data was omited in our training set. Our primary objective is to expand motion synthesis to underrepresented non-human species within a unified framework. To this end, we used the Truebones Zoo dataset, which consists of ~1K motion clips over 70 non-human object categories. In contrast, existing human motion datasets often comprise tens of thousands to millions of samples. Incorporating human motion would introduce a significant imbalance, likely skewing the model’s capacity toward human motion and limiting its ability to learn meaningful representations for underrepresented yet a lot more diverse categories.
>
> We will include a qualitative example and further discussion of this limitation in the revision. In future work, we plan to address this by scaling to larger datasets, such as Objaverse-XL, to construct a more balanced corpus of human and non-human motions. This would enable training a unified model capable of generalizing across a broader spectrum of skeletal structures and motion styles.
>
> ### Q5: How to keep the motion quality of the augmented rigs.
> Yes, we retarget the original motion to the augmented skeletons, using Blender’s retargeting pipeline.
>
> To ensure motion quality, we carefully designed the augmentation process to minimize disruption to motion dynamics. Changes like rest pose adjustments and bone subdivision have limited impact, while more sensitive bone length adjustment and bone erasing were applied conservatively.
> - Length adjustments were restricted to 0.8×–1.2× of the original and applied symmetrically when symmetry existed.
> - Bone erasing was limited to distal appendages (toes, head tips, tail ends) and redundant spine/root bones.
>
> Under these constraints, we visually verified ~10K augmented results and found them suitable for training, contributing to improved generalization across diverse skeletons.
>
> We acknowledge that these details were not fully described in the original submission, and we will include a more detailed explanation in the revision to clarify the design choices and safeguards used in the rig augmentation process. To further ensure plausibility, exploring quantitative checks (e.g., joint velocity, momentum shifts, foot contact, end-effector deviations), followed by rejection sampling could be a promising direction. Another promising direction is two-stage training: pretraining on augmented rigs to improve generalization and bone manipulation, then fine-tuning on physically grounded data for higher realism.
>
> ### Q6: Further ablation study on TreePE vs RestPE
> We conducted an ablation study to isolate the effects of PEs. This experiment was performed under the same setting as Table 1 in the paper.
>
> | RestPE | TreePE | Rig Aug. | Train |  |  | Test |  |  | Test+ |  |  |
> |:---:|:---:|:---:|:---:|:---:|:---:|:---:|:---:|:---:|:---:|:---:|:---:|
> |  |  |  | FID (↓) | R@1 (↑) | Align. (↑) | FID (↓) | R@1 (↑) | Align. (↑) | FID (↓) | R@1 (↑) | Align. (↑) |
> | X | X | O | 0.98 | 0.33 | 0.66 | 2.27 | 0.26 | 0.68 | 0.86 | 0.39 | 0.74 |
> | X | O | O | 0.15 | 0.85 | 0.90 | 1.17 | 0.53 | 0.83 | 0.48 | 0.58 | 0.84 |
> | O | X | O | 0.02 | 0.95 | 0.95 | 0.69 | 0.59 | 0.87 | 0.42 | 0.66 | 0.89 |
> | O | O | O | 0.01 | 0.97 | 0.97 | 0.68 | 0.60 | 0.89 | 0.26 | 0.67 | 0.93 |
>
> The results confirm that both are beneficial: RestPE has a stronger impact, but the best performance is achieved when both are used. We believe the strong performance of RestPE alone is partly due to the model's ability to implicitly infer parent-child relationships from relative offsets and bone lengths.

---

> > ### Comment · Reviewer_2SeY · 2025-04-08
> >
> > Thanks for the further explanations. The authors solved most of my questions. However, the method still suffers from poor generalizations. The author could select the proper amount of human data from the Truebones zoo dataset to verify whether the framework works for humanoid or not. Thus, I decided to keep my original score.

---

> > > ### Author Response · Authors · 2025-04-08
> > >
> > > We sincerely thank the reviewer for the follow-up comment and thoughtful suggestions.
> > >
> > > We would like to reiterate the primary objective of our work: to push the boundary of motion synthesis toward more diverse object categories that differ significantly in skeletal structure. Unlike prior works that focus primarily on a limited set of object types representable with a single fixed skeletal topology, our goal is to explore whether a unified framework can accommodate the complexity and variability of morphologically diverse categories.
> > >
> > > That said, while the Truebones dataset is a highly valuable resource due to its diversity and quality, it remains relatively small in scale—especially when compared to the datasets used to train foundation models in language or vision domains, where zero-shot inference is often the primary focus of evaluation. Given this limitation, we believe it is unreasonable to expect strong zero-shot capabilities in our setting, akin to those demonstrated by much larger models. Moreover, even with the inclusion of a small curated set of human skeletons, the broader challenge of generalizing to truly underrepresented object categories likely remains unresolved.
> > >
> > > Nevertheless, we believe our experiments on the Truebones Zoo dataset offer promising evidence that our method can effectively handle a wide range of skeletons and motion styles—and even generalize to previously unseen categories. We view this as a meaningful step toward broader motion synthesis for open-vocabulary objects, extending well beyond the limited subset of categories addressed in prior work or covered in our current study.
> > >
> > > We hope this response further clarifies the motivation and contributions of our work, and we sincerely appreciate the reviewer’s constructive feedback and the opportunity to elaborate on our approach.

---

### Official Review · Reviewer_cRx3 · 2025-03-13

**Overall Recommendation:** 3

**Summary:**

This work presents a unified framework for motion synthesis across a diverse range of objects with varying skeletal structures and rest poses. To generate training data, the authors augment the Truebones Zoo dataset by modifying skeletal structures and rest poses , and providing textual descriptions at multiple levels of detail.

The diffusion model is built upon DiT, where the technical contributions lie in the new TreePE and RestPE layers introduced to accommodate arbitrary skeletal topologies and rest poses. Additionally, a two-stage training strategy is implemented to address computational resource constraints and data limitations.

Based on the reported quantitative and qualitative results, the proposed method demonstrates strong performance compared to baselines and exhibits generalizability to novel skeletons and motion descriptions.

## update after rebuttal

After reading the rebuttal, I will maintain my score as weak accept. I do feel the solution proposed by the authors is interesting, and with the release of the augmented dataset, it should have border impact for subsequent works in related areas such as artificial agents and robotics.

**Claims And Evidence:**

- The necessity of the proposed positional encoding layers and data augmentation techniques is validated through ablation studies.
- The method's generalizability to diverse rigs and motion descriptions is demonstrated through both quantitative and qualitative experiments.

**Essential References Not Discussed:**

References are fairly enough for me.

**Experimental Designs Or Analyses:**

- The ablation study examines the necessity of positional encodings and rig augmentation. It looks good to me.
- The quantitative comparisons in Figure 4 compares the proposed model design with other baseline architectures on text-to-motion synthesis. It's sound to me, with only one small question: why does the simple retargeting method perform better than other baselines? Any insights here?
- Motion synthesis on novel objects and skeletons are qualitatively evaluated on different species, which are reasonable to me.
- Long motion sequence generation is achieved by conditioning on sequential descriptions with a sliding window. For this experiment, I would like to see more detailed descriptions like how the weighted blending is performed. I can understand the idea in general, but I'm a bit skeptical since I don't think naively blending the joint rotations can always ensure a natural and smooth transition. It's more like a motion in-betweening problem, so more detailed description and analysis are welcome.
- Generating motions with multi-level descriptions is sound to me.

**Methods And Evaluation Criteria:**

The proposed method and evaluation are reasonable.

**Other Comments Or Suggestions:**

- As the two-stage training stategy is specially designed for computational resource constraints and data limitations, it would be helpful to include details on the hardware specs used for model training as well as the overall training time.

**Other Strengths And Weaknesses:**

[Strengths]

- The paper proposes a motion synthesis framework capable of handling skeletal topologies and rest poses, covering a wide range of objects.
- The authors augment the Truebones Zoo dataset to include more skeletal variations and high-quality text descriptions. The release of this dataset should faciliate research in related areas.
- Both quantitative results and qualitative evaluations demonstrate that the proposed method surpasses baselines in terms of generation quality.
- The paper is well-written, technically correct and good quality.

[Weaknesses]

- No global translation. Due to the model design, all the generated motions are pinned at the origin without global translation of the root joint. It would be beneficial to include discussions on how to generate the global translation, expecially for those non-humanoid skeletons.

**Questions For Authors:**

n/a

**Relation To Broader Scientific Literature:**

This paper proposes a motion synthesis framework for a broad range of objects, extending previous motion diffusion models beyond humanoid applications. In general, this approach has potential applications in areas such as artificial agents and robotics, enabling motion synthesis for non-humanoid robots.

**Theoretical Claims:**

n/a

---

> ### Author Rebuttal · Authors · 2025-03-31
>
> We appreciate the reviewer for raising important points and providing constructive feedback.
>
> ### Q1: Why do the simple retargeting method perform better than other baselines?
>
> Data-driven learning-based methods each have limitations in generalization or controllability:
> - GPT-Caption is trained on captions automatically extracted from rendering images of motions, which tend to be noisy or ambiguous. This weakens the model’s ability to learn reliable text-motion alignment, resulting in limited text-driven controllability.
> - Single-Object MDM is trained on motion from a single object category and tends to overfit, making it difficult to generalize beyond the training distribution.
> - w/o Rig Aug is trained on high-quality captions and diverse objects, but without rig augmentation, the model lacks sufficient exposure to structural variation, limiting its cross-embodiment transfer capability, as our response to Q3 of Reviewer 4SvT.
>
> In contrast, retargeting directly copies bone transformations from source motion. This naturally boosts novelty and, when source and target skeletons are similar, leads to higher coverage. That said, retargeting can't generate motion from text and depends on existing motions and structural similarity.
>
> ### Q2: More description and analysis for long-motion synthesis.
>
> We fully agree that naively blending joint rotations, regardless of the representation used (e.g., euler angles, quaternions, rotation matrices, or even continuous 6D representations), does not always guarantee a smooth or physically plausible transition, due to the discontinuous and nonlinear nature of rotational spaces [1]. We also acknowledge that this is closely related to motion in-betweening, which typically requires dedicated interpolation strategies or learned dynamics.
>
> To generate long motions, we prepare $B$ consecutive text descriptions, each guiding a motion segment of length $F$, resulting in a long sequence of $B \times F$ frames. Before denoising, we reorganize the sequence into $B$ overlapping segments. Specifically, the $b$-th segment spans frames from $(b - 1)F - (b-1)W + 1$ to $bF - (b - 1)W$ in the original sequence, introducing overlaps of $W$ frames with both the $(b - 1)$-th and $(b + 1)$-th segments. The first segment ($b = 1$) is an exception and simply takes the first $F$ frames, i.e., frames $1$ to $F$. This reorganization results in duplicated content within the overlapping regions. It’s worth noting that, as a result, the final segment ends at frame $BF - (B - 1)W$, and any frames beyond that (i.e., from $BF - (B - 1)W + 1$ onward) are ignored during sampling. Each divided segment is then denoised independently, conditioned on its respective caption. After denoising, the segments are concatenated to reconstruct the full sequence, while the overlapping regions are blended by simple averaging to resolve duplication.
>
> While our implementation uses uniform blending (i.e., averaging), more sophisticated strategies—such as linear ramps or cosine-weighted curves—can also be applied. Despite its simplicity, we empirically found that uniform averaging produces smooth, coherent transitions, especially when the overlap size $W$ is moderate (e.g., $W = 5$) and the text prompts transition naturally between segments.
>
> As also noted by Reviewer 4SvT, we evaluated long motion quality via semantic fidelity and smoothness, comparing it to short sequences ($F=90$) used during training, to assess whether quality degrades over time. Please find our response to Q2 of Reviewer 4SvT for details.
>
> References:
>
> [1] Zhou et al., On the Continuity of Rotation Representations in Neural Networks, CVPR 2019.
>
> ### Q3: About global translation.
>
> In our current setup, we model only joint-wise euler angle rotations and omit global translation of the root joint. However, this was a design choice made for simplicity, not a fundamental limitation of the model.
>
> The model operates on input-output tensors of shape $F \times J \times D$, where $D = 3$ corresponds to joint rotations. This can be naturally extended to include additional joint-level features—such as global root translation or relative translations for soft-constrained joints—by increasing $D$ (e.g., $D = D_\text{rot} + D_\text{trans}$), or further expanded to incorporate physically informative signals like joint velocities, foot contact indicators, or positions to enhance temporal coherence, realism, and physical plausibility.
>
> We will include a discussion of this direction in the revised manuscript.
>
> ### Q4: About computational resources.
>
> All models were trained on a Linux system with either an NVIDIA RTX 48GB A6000 or 40GB A100 GPU.
> - Pose Diffusion Model: \~29GB VRAM, batch size 512, 400K iterations (\~30 hours)
> - Motion Diffusion Model: \~38GB VRAM, batch size 4, sequence length 90, 1M iterations (\~4 days)
>
> We will include this information in the revised manuscript.

---

### Official Review · Reviewer_4SvT · 2025-03-16

**Overall Recommendation:** 3

**Summary:**

This work presents a major advancement in text-driven motion synthesis for large-vocabulary objects with heterogeneous skeletal structures. By augmenting datasets, introducing novel rig adaptation techniques, and extending motion diffusion models, the authors enable realistic motion synthesis for both seen and unseen objects. This framework lays a strong foundation for animation, gaming, virtual reality, and robotics applications.

**Claims And Evidence:**

1. Claim: The proposed method significantly outperforms all existing approaches in text-to-motion generation.
Partially supported, but lacks direct baseline comparisons.
While the model is tested against ablated versions (e.g., without rig augmentation or using GPT-generated captions), there is no direct comparison with existing SOTA models like OmniMotion-GPT or SinMDM. The paper would benefit from side-by-side quantitative evaluations against prior text-to-motion models using the same datasets.

2. Claim: The framework enables high-fidelity, temporally coherent long-motion synthesis.
Limited empirical validation.
Figure 7 shows generated longer sequences, but there are no explicit temporal consistency metrics (e.g., FID over extended motion sequences, smoothness scores).
The paper could strengthen this claim by adding quantitative evaluations of motion continuity and stability over time.

**Essential References Not Discussed:**

N. A

**Ethical Review Concerns:**

N.A

**Experimental Designs Or Analyses:**

N. A

**Methods And Evaluation Criteria:**

N.A

**Other Comments Or Suggestions:**

N.A

**Other Strengths And Weaknesses:**

N.A

**Questions For Authors:**

N.A

**Relation To Broader Scientific Literature:**

N. A

**Theoretical Claims:**

No Theoretical Bound for Generalization to Novel Skeletons:
The claim that the model generalizes well to unseen skeletons (Section 7.1) is based on empirical evidence, but there is no theoretical proof explaining why TreePE and RestPE should generalize across arbitrary skeletal configurations.

---

> ### Author Rebuttal · Authors · 2025-03-31
>
> We sincerely appreciate the reviewer’s valuable time and feedback.
>
> ### Q1: Comparisons with SOTA models like OmniMotion-GPT or SinMDM.
>
> We agree that comparisons to SOTA models are valuable. However, OmniMotion-GPT and SinMDM address different goals and operate under different assumptions, making direct, meaningful comparisons less feasible.
>
> OmniMotion-GPT leverages human motion priors for text-driven motion synthesis of SMAL-based quadrupeds. It relies on:
> - Predefined joint correspondences between humans and animals, supporting only fixed, four-limbed skeletons (e.g., quadrupeds), and failing on animals with non-analogous structures (e.g., snakes, fish, insects).
> - Text input alone is insufficient: relevant human motion is required at inference (e.g., animating “a lion pushing a box” needs a human pushing motion), limiting text-only synthesis.
> - Since our dataset (Truebones Zoo) contains a wide variety of non-human skeletons for which such paired human motion does not exist, applying OmniMotion-GPT in this context is infeasible.
>
> SinMDM, while based on MDM like ours, differs fundamentally:
> - It aims to discover and recombine submotions (i.e., motion motifs) from a single clip using local attention to increase the diversity within a single motion, rather than focusing on text-driven synthesis or spanning across object categories.
> - It assumes a fixed skeleton and rest pose, making it incompatible with varied or unseen structures, and is not text-conditioned.
> - To adapt SinMDM for our setting, it would require substantial changes: introducing text conditioning and training a separate model per object type. In that case, the closest equivalent in our experiments is Single-Object MDM, which we already include as a baseline.
>
> That said, we view SinMDM as orthogonal and potentially complementary to our work. While SinMDM enhances diversity within a fixed structure and motion, our model focuses on text-driven synthesis and cross-categories modeling. Combining these directions would be a promising avenue for future research.
>
> ### Q2: Evaluation on synthesized long motions.
>
> We thank the reviewer for the suggestion. We evaluated both semantic consistency and temporal smoothness in long-form motion with respect to the original short-form motion.
>
> To assess semantic fidelity, we conducted a retrieval test using the pretrained pose encoder:
> - Using 10 captions from Sec. 7.2, we generated 100 long motions (900 frames each = 10 × 90-frame segments), with shuffled caption order. These were split into 1000 90-frame segments.
> - Separately, we generated 100 90-frame reference clips per caption (1000 total).
> - For each segment, we retrieved the top match from the reference set using cosine similarity. A correct match retrieved a clip from the same caption.
>
> This yielded 95.6% top-1 accuracy, indicating that segments remain semantically aligned after blending.
>
> To assess temporal smoothness, we measured joint velocity norm (lower = smoother) and Motion Stability Index (MSI [1,2]; higher = smoother) over 5-frame clips sampled from three conditions, using the same generated and reference sequences as above:
> - Upper Bound: from random mid-segments within reference clips — velocity = 0.094, MSI = 79,602
> - Ours: from overlapping segments with blending in long-form generated motion — velocity = 0.188, MSI = 26,716
> - Baseline: from hard concatenation of independently generated clips — velocity = 0.444, MSI = 3.11
>
> These results show that our method significantly reduces boundary discontinuity compared to naive concatenation, while preserving motion dynamics close to intra-clip smoothness. We will include these results in the revised manuscript.
>
> References:
>
> [1] Ling et al., StableFace: Analyzing and Improving Motion Stability for Talking Face Generation, ECCV 2022.
>
> [2] Kim et al., Audio-driven Talking Face Generation with Stabilized Synchronization Loss, CVPR 2023.
>
> ### Q3: Generalization to unseen skeletons
>
> TreePE and RestPE are introduced to make the model skeleton-aware, aiding learning, while rig augmentation is designed to drive generalization.
>
> Our intuition behind rig augmentation is simple: First, by constructing and showing the model a distribution of skeletons that share the same motion and caption, we encourage it to focus on the motion’s semantic meaning rather than its specific structural form. In addition, by displaying a variety of skeleton topologies and rest poses for each motion, we aim to teach the model how to manipulate bones to generate the motion. Moreover, as these distributions expand through augmentation, they may begin to overlap across different object classes, which we believe acts as a bridge and helps the model transfer motion patterns more effectively to novel skeletons.
>
> We’ll clarify this in the revision—our intent was for PE to support learning via structural awareness, and for rig augmentation to promote generalization by encouraging structural invariance.

---

### Decision · Program_Chairs · 2025-05-01

**Decision:**

Accept (poster)

**Comment:**

The work received very supportive reviews from the reviewers even prior to the rebuttal. They acknowledged that the manuscript presents a significant advancement in motion synthesis. All four reviewers agreed that the paper is a valuable contribution. The authors provided a strong rebuttal and successfully addressed the remaining concerns. Therefore, the AC recommends acceptance. Congrats!